# CK2-induced cooperation of HHEX with the YAP-TEAD4 complex promotes colorectal tumorigenesis

Yuegui Guo[1,2,5], Zhehui Zhu[1,2,3,5], Zhenyu Huang[1,2], Long Cui [1,2], Wei Yu[3], Wanjin Hong [4], Zhaocai Zhou [3] ✉, Peng Du [1,2] ✉ & Chen-Ying Liu [1,2] ✉

Dysregulation of Hippo pathway leads to hyperactivation of YAP-TEAD transcriptional complex in various cancers, including colorectal cancer (CRC). In this study, we observed that HHEX (Hematopoietically expressed homeobox) may enhance transcription activity of the YAP-TEAD complex. HHEX associates with and stabilizes the YAP-TEAD complex on the regulatory genomic loci to coregulate the expression of a group of YAP/TEAD target genes. Also, HHEX may indirectly regulate these target genes by controlling YAP/TAZ expression. Importantly, HHEX is required for the pro-tumorigenic effects of YAP during CRC progression. In response to serum stimulation, CK2 (Casein Kinase 2) phosphorylates HHEX and enhances its interaction with TEAD4. A CK2 inhibitor CX-4945 diminishes the interaction between HHEX and TEAD4, leading to decreased expression of YAP/TEAD target genes. CX-4945 synergizes the antitumor activity of YAP-TEAD inhibitors verteporfin and Super-TDU. Elevated expression of HHEX is correlated with hyperactivation of YAP/TEAD and associated with poor prognosis of CRC patients. Overall, our study identifies HHEX as a positive modulator of YAP/TEAD to promote colorectal tumorigenesis, providing a new therapeutic strategy for targeting YAP/TEAD in CRC.

The Hippo pathway is an evolutionarily conserved signaling pathway regulating organ size, tissue development, and homeostasis[1]. Dysregulation of the Hippo pathway occurs in multiple cancers, including colorectal cancer (CRC)[1,2]. Yes-associated protein (YAP) and Transcriptional coactivator with PDZ-binding motif (TAZ) act as the main downstream effectors of the Hippo pathway. As transcriptional coactivators, YAP/TAZ can bind with various transcription factors to control the expression of downstream target genes. The TEAD family members (TEAD1/2/3/4) are the main transcription factors mediating the oncogenic function of YAP/TAZ[3]. Transcription factors (TFs) recruit multiple cofactors and often cooperate with other TFs to synergistically regulate gene transcription[4]. For example, the transcription factor AP1 has been reported to interact with YAP-TEAD to form a complex, which cooperatively activates target gene expression to drive YAP-dependent oncogenic growth in breast and skin cancer[5]. Recently, we discovered that IRF3, a central transcription factor in antiviral innate immunity, acts as an agonist of YAP-TEAD in gastric cancer and that pharmacological inhibition of IRF3 by amlexanox suppresses YAP-driven gastric tumor growth[6]. However, the mechanism underlying the regulation of YAP/TEAD transcriptional activity and the related synergistic regulators of YAP/TEAD in colorectal cancer remain poorly understood.

Hematopoietically expressed homeobox (HHEX or HEX), also called proline-rich homeodomain (PRH), is a transcription factor with a

[1]Department of Colorectal and Anal Surgery, Xinhua Hospital, Shanghai Jiao Tong University School of Medicine, Shanghai 200092, China. [2]Shanghai Colorectal Cancer Research Center, Shanghai 200092, China. [3]State Key Laboratory of Genetic Engineering, School of Life Sciences, Zhongshan Hospital, Fudan University, Shanghai 200438, China. [4]Institute of Molecular and Cell Biology, Agency for Science, Technology and Research (A*STAR), 61, Biopolis Drive, Proteos, Singapore 138673, Singapore. [5]These authors contributed equally: Yuegui Guo, Zhehui Zhu. ✉e-mail: zhouzhaocai@fudan.edu.cn; dupeng@xinhuamed.com.cn; liuchenying@xinhuamed.com.cn

homeodomain for DNA binding[7]. Although HHEX contains a C-terminal acidic activation domain and has been reported to transcriptionally activate the bile transporter gene *NTCP*, studies have indicated that HHEX acts as a major transcriptional repressor by recruiting TLE corepressor proteins to a subset of target promoters[7–12]. In addition, the homeodomain of HHEX can be phosphorylated by Casein Kinase 2 (CK2), which suppresses the DNA-binding activity and transcriptional repressor function of HHEX[13,14]. In addition to its crucial role in hematopoietic and vascular development, HHEX has also been implicated in the pathology of leukemias and various solid tumors in a cell context-dependent manner[15]. For example, HHEX functions as an oncogene or a tumor suppressor in different subtypes of acute myeloid leukemia (AML) and CML[15]. In solid tumors, HHEX has been found to act as a suppressor of cell migration and cell proliferation in breast, liver, and prostate cancers[15]. Recently, HHEX has been shown to be an oncogenic driver in cholangiocarcinoma through transcriptionally activating the Wnt and NOTCH pathways[16].

In this study, we report that HHEX is a positive regulator of the YAP-TEAD4 complex in CRC. HHEX physically interacts with YAP and TEAD4 through different domains to enhance the transcriptional activity of the YAP-TEAD4 complex. HHEX is upregulated in human CRC and required for the oncogenic function of YAP/TAZ in CRC. Upregulation of HHEX is positively correlated with the transcriptional activity of YAP in CRC, and is associated with poor clinical outcomes in CRC patients. Importantly, inhibition of CK2 decreases the interaction of HHEX with TEAD4 and synergistically suppresses CRC growth in combination with the YAP-TEAD inhibitor verteporfin. Thus, our study reveals a new cooperative regulatory mechanism of YAP/TEAD4 transcriptional activity by HHEX, highlighting the therapeutic potential of targeting the HHEX-TEAD4 interaction in CRC.

## Results

### HHEX interacts with the YAP-TEAD complex

Previously, we revealed that TEAD4 was overexpressed in CRC and promoted tumor metastasis in a YAP-independent manner[17]. To identify new transcriptional coregulators of TEAD4 in CRC, we generated HCT-116 cells stably expressing FLAG-TEAD4 WT and performed immunoprecipitation-mass spectrometry (IP-MS/MS) analysis. Several transcription factors known to interact with TEAD4, including YAP/TAZ, VGLL4, and TCF7L2, were identified (Fig. 1a). In addition, we noted that HHEX, a member of the homeobox family of transcription factors, was also among the top candidates for binding partners of TEAD4 (Fig. 1a). Subsequently, we confirmed the interaction between TEAD4 and HHEX by coimmunoprecipitation (co-IP) in HEK-293T cells overexpressing FLAG-TEAD4 and HA-HHEX (Figs. 1b, S1a). In addition, DNase treatment did not decrease the interaction of TEAD4 with HHEX, indicating the DNA-independent nature of the interaction (Fig. 1c). We also found that HHEX interacted with TEAD1/2/3 (Figure S1b), suggesting that HHEX acts as a general coregulator of TEAD family transcription factors. Next, we tested whether HHEX can interact with the TEAD coactivator YAP/TAZ. The results of the co-IP assay showed that both YAP and TAZ interacted with HHEX in HEK-293T cells (Figs. 1d, S1c). Moreover, overexpression of HHEX moderately enhanced the interactions between YAP and TEADs in HEK-293T cells (Fig. 1e). To further confirm the role of HHEX in enhancing YAP-TEAD interactions, we constructed HCT-116 cell lines stably expressing HHEX or shRNA targeting *HHEX* (Figure S1d). Consistent with the above findings, overexpression of HHEX promoted interactions between YAP and TEADs while knockdown of *HHEX* decreased the interaction in HCT-116 cells (Fig. 1f). Furthermore, both endogenous co-IP and proximity ligation assay (PLA) confirmed the interaction between endogenous HHEX and endogenous TEAD4 or YAP in HCT-116 cells (Figs. 1g, h and S1e). Knockdown of *HHEX* reduced the nuclear PLA signals in HCT-116 cells (Fig. 1h). Immunofluorescence staining also showed colocalization of HHEX and YAP/TAZ in the nucleus

(Figure S1f). Collectively, these data demonstrate that the transcription factor HHEX is a new interacting partner of TEAD4 and YAP in CRC and that HHEX stabilizes the YAP-TEAD complex.

### Distinct domains of HHEX mediate its interaction with YAP and TEAD4

HHEX is well known as a transcriptional repressor through binding to the corepressor protein TLE-1[18]. To explore the nature of the interaction between HHEX and YAP/TEAD in the context of HHEX's traditional cofactors, we generated several HHEX mutants, including an F32E mutant unable to bind TLE-1[18], an N187A mutant defective for DNA binding[19], and an L23A/L24A mutant unable to bind eIF4E[20]. We then tested the effect of these mutations on the interaction of HHEX with YAP/TEADs. The co-IP assay showed that the N187A and L23A/L24A mutations in HHEX did not affect its interaction with YAP, while the F32E mutation in HHEX moderately enhanced its interaction with YAP compared to that of wild-type HHEX (Figure S2a, b). Subsequently, we tested whether YAP forms a complex with HHEX and TLE1. We found that overexpression of YAP did not promote the interaction between HHEX and TLE-1 and that YAP could not interact with TLE-1, indicating that different HHEX pools form protein complexes with YAP and TLE-1 (Figure S2c, d).

Next, we investigated the interactions of wild-type and mutant HHEX with TEAD4. The TLE-1 binding-defective F32E mutant and the eIF4E binding-defective L23A/L24A mutant of HHEX retained the ability to bind TEAD4 (Figure S2e). However, the N187A mutation in HHEX, which rendered HHEX unable to bind DNA, almost completely abolished the interaction with TEAD4 (Fig. 1i) but did not affect the ability of HHEX to bind YAP (Figure S2a). These results indicated that HHEX interacts with YAP and TEAD4 through distinct domains or interfaces. In support of this hypothesis, neither the S94A mutation in YAP, which rendered it defective in binding TEAD, nor the Y429H mutation in TEAD4, which rendered it defective in binding YAP, significantly altered the interactions between HHEX and YAP/TEAD4 (Figure S2f, g). The HHEX protein consists of three domains: an N-terminal proline-rich domain, an acidic C-terminal region, and a homeodomain for DNA binding between the N- and C-terminal domains[7]. We then sought to determine which domain of HHEX is required for its interaction with YAP. The results showed that a truncated form of HHEX lacking the N-terminal proline-rich domain could no longer interact with YAP (Fig. 1j), indicating that the N-terminal proline-rich domain of HHEX mediates its interaction with YAP.

By generating various N-terminal truncation mutants of TEAD4, we found that the N-terminal TEA domain of TEAD4 was indispensable for binding HHEX (Figure S3a). In particular, deletion of the DNA-binding interface region significantly disrupted the interaction of TEAD4 with HHEX (Figure S3a). Moreover, individual mutation of the interface residues V42 and W43 in TEAD4 also significantly attenuated its interaction with HHEX (Figure S3b). Given that both V42 and W43 in TEAD4 are required for its DNA binding, we tested the ability of another TEAD4 mutant, S100A—a TEA domain mutant unable to bind DNA[21]—to interact with HHEX. We found that the S100A mutation in TEAD4 mildly diminished its interaction with HHEX, demonstrating that the DNA-binding domain of TEAD4 mediates the interaction between TEAD4 and HHEX (Figure S3c). Taken together, these results indicate that HHEX interacts via distinct domains with both YAP and TEAD to stabilize the YAP-TEAD complex.

### HHEX promotes the transcriptional activity of the YAP-TEAD complex for tumorigenesis

Next, we sought to determine whether HHEX can regulate the transcriptional activity of YAP/TEAD. The luciferase reporter assay results showed that co-expression of HHEX with either wild-type YAP/TAZ or their constitutively active form (YAP^SSA/TAZ^4SA) significantly promoted the transactivation of TEAD in a dose-dependent manner (Figs. 2a,

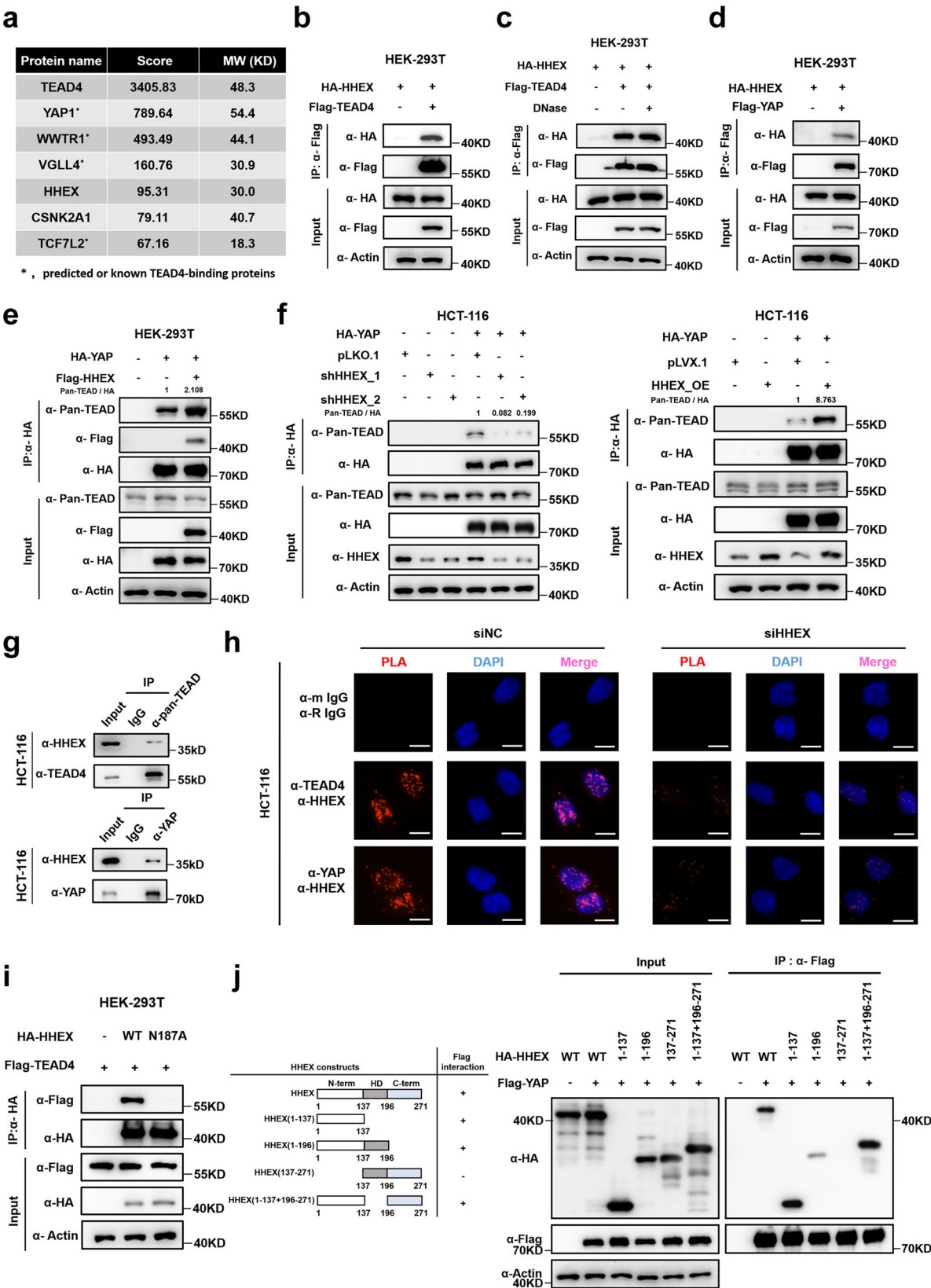

S4a). Interestingly, the F32E mutant, which shows enhanced interaction with YAP compared to the WT HHEX, induced a 2 fold higher activation of the TEAD luciferase activity than the WT HHEX (Figure S4b). Notably, the N187A mutant and N-terminal truncation mutant of HHEX, both are known to have attenuated transcriptional activity, largely retained the ability to promote TEAD transactivation (Fig. 2b,

c), indicating that the interactions of HHEX with both YAP and TEAD but not the transcriptional activity of HHEX itself were required for full activation of the YAP-TEAD complex by HHEX. To identify the target genes co-regulated by HHEX and YAP/TEAD, we performed RNA-seq analysis of HCT-116 cells with individual knockdown of *HHEX, YAP/TAZ* and *TEAD1/2/3/4* (Supplementary Data 1). The Venn diagram showed

**Fig. 1 | HHEX interacts with YAP/TEAD4 complex. a** Mass spectrometry (MS) identification of TEAD4-interacting proteins in HCT-116 cells. **b** Co-IP of exogenous HA-HHEX and FLAG-TEAD4 in HEK-293T cells. **c** Co-IP of exogenous HA-HHEX and FLAG-TEAD4 in HEK-293T cells. The cell lysate was treated with or without Benzonase before immunoprecipitation. **d** Co-IP of exogenous HA-HHEX and FLAG-YAP in HEK-293T cells. **e** HHEX increased the interaction between YAP and TEAD. Semiendogenous co-IP of exogenous HA-YAP and endogenous TEADs in HEK-293T cells with or without overexpression of FLAG-HHEX. **f** HHEX enhanced the interaction between YAP and TEAD in CRC cells. Semiendogenous co-IP of exogenous HA-YAP and endogenous TEADs in HCT-116 cells with *HHEX* knockdown or overexpression. **g** Endogenous co-IP of YAP/TEAD4 and HHEX in HCT-116 cells. **h** In situ PLA (red signal) of YAP/TEAD4 and HHEX was performed with anti-YAP/TEAD4 and anti-HHEX antibodies in the *HHEX* knockdown and control HCT-116 cells. Scale bars, 10 μm. **i** The HHEX-N187A mutant abolished the interaction between HHEX and TEAD4. Co-IP of exogenous HA-HHEX WT/N187A and FLAG-TEAD4 in HEK-293T cells. **j** The co-IP assay confirmed the interaction of the N-terminal domain of HHEX with YAP. A schematic showing the protein structure of the full-length and truncated HHEX proteins (left). These data (**b**–**j**) are representative of 3 independent experiments. Source data are provided as a Source data file.

that 28 genes were co-downregulated by individual knockdown of *HHEX, YAP/TAZ*, and *TEAD1/2/3/4* (fold change > 2, $p < 0.05$) and a hypergeometric test also showed statistically overlap of the differentially downregulated genes (*siTEAD* vs *siHHEX*, $p = 3.60e^{-54}$; *siYAP/TAZ* vs *siHHEX*, $p = 1.19e^{-57}$) (Fig. 2d). Moreover, gene set enrichment analysis (GSEA) of the RNA profile data for HCT-116 cells with *HHEX* knockdown revealed that the YAP target genes were downregulated (Fig. 2e). qPCR analysis of the well-known target genes of YAP/TEAD, such as *CTGF, CYR61, AXL, ANKRD1*, and *LATS2*, further confirmed the GSEA results showing downregulation of YAP/TEAD transcriptional activity in HHEX-deficient HCT-116 cells (Figs. 2f, S4c). Similar results were also observed in *HHEX* knockdown SW-480 cells (Figure S4d, S4e). It is worth noting that both mRNA and protein levels of *YAP* and *TAZ* were downregulated in the *HHEX* knockdown HCT-116 cells (Figure S4f, S4g). Intriguingly, overexpression of HHEX increased the protein levels but not the mRNA levels of *YAP* and *TAZ* in the HCT-116 cells (Figure S4f, S4g). These data indicated that HHEX could regulate the transcriptional activity of YAP/TEAD through multiple mechanisms, in addition to complexing with YAP/TEAD. In this study, we focused on the cooperative regulatory mechanism of YAP/TEAD4 transcriptional activity by HHEX.

To gain the genomic binding profile of HHEX in CRC cells, we performed the ChIP-seq analysis of HHEX in HCT-116 cells (Fig. 2g). We also annotated the TEAD4 binding sites by taking advantage of the public ChIP-seq dataset of TEAD4 in HCT-116 from ENCODE. 6876 peaks with highly credibility (IDR < 0.02) were extracted from the ENCODE TEAD4 ChIP-seq dataset (Fig. 2g and Supplementary Data 2). The ChIP-seq analysis of HHEX in HCT-116 cells identified 4697 peaks targeting 4381 genes (Supplementary Data 2). 767 genes were co-regulated by TEAD4 and HHEX in HCT-116 cells (Fig. 2h). Similar results were observed in HepG2 cells based on the publicly available ENCODE ChIP-sequence datasets of TEAD4 and GFP-HHEX (Figure S4h, S4i). The TEAD4/HHEX co-bound genes include the classic target genes of YAP/TEAD (*CTGF, CYR61, LATS2, AMOTL2*) (Fig. 2i). The ChIP-qPCR results further confirmed that HHEX bound to the promoter regions of *CTGF* and *CYR61* and to the enhancer region of *ANKRD1*, which were bound by TEAD (Figs. 2j, S4j). Furthermore, knockdown of *TEAD1/3/4* significantly diminished the recruitment of HHEX to the promoter region of these classical TEAD direct target genes but not the previously reported HHEX target gene *VEGFA*[10,22,23] (Figs. 2j, S4k).

Hyperactivation of YAP/TEAD is frequently observed in cancer and contributes to tumorigenesis[2,24,25]. Thus, we further sought to determine whether HHEX mediates the pro-oncogenic role of YAP/TEAD in CRC cells. To this end, we examined whether the YAP overexpression-induced oncogenic phenotype can be attenuated by knockdown of *HHEX* in HCT-116 cells. We observed that overexpression of both WT YAP and the constitutively active YAP-5SA mutant significantly enhanced xenograft tumor growth and the mRNA expression levels of the classical YAP/TEAD direct target genes (Figs. 2k, l and S5a–f). As expected, knockdown of *HHEX* in HCT-116 cells expressing YAP WT and YAP-5SA dramatically attenuated the pro-oncogenic effects of YAP WT and YAP-5SA (Figs. 2k, l and S5a–f). Taken together, these results indicate that HHEX is essential for the pro-

oncogenic activity of the YAP in CRC and suggest that HHEX may promote colorectal tumorigenesis via YAP/TEAD.

## HHEX is a pro-tumorigenic gene in CRC

Previously, HHEX was reported to act as a tumor suppressor in breast cancer and prostate cancer[12,26,27]. However, the finding that HHEX was required for the pro-oncogenic role of YAP/TEAD in CRC cells prompted us to explore the potential pro-tumorigenic role of HHEX in CRC. By using CCK-8 and colony formation assays, we observed that knockdown of *HHEX* inhibited the proliferation and colony formation of HCT-116 and SW-480 CRC cells, whereas overexpression of HHEX elicited the opposite effects (Fig. 3a–d). To verify the role of HHEX in CRC in vivo, we performed a xenograft assay by subcutaneously injecting HCT-116 cells into nude mice. Consistent with the above findings, knockdown of *HHEX* in HCT-116 cells dramatically inhibited tumor growth, while xenograft tumors derived from HCT-116 cells overexpressing HHEX were significantly larger than tumors derived from control HCT-116 cells (Fig. 3e). Ki67 staining of the xenograft tumors further showed that tumors derived from *HHEX* knockdown cells contained fewer Ki67-positive tumor cells and that overexpression of HHEX increased the number of Ki67-positive cells in the xenograft tumors (Fig. 3f). Collectively, our data indicate that HHEX plays a pro-tumorigenic role in CRC, probably by activating the transcriptional activity of the YAP-TEAD complex.

## Knockout of HHEX attenuates colitis-associated colorectal cancer progression

To assess the function of HHEX in tumorigenesis, we specifically deleted *Hhex* in intestinal epithelial cells by generating *Villin-Hhex*[f/f] mice (Fig. 4a). First, we confirmed the knockout efficiency of *Hhex* (Fig. 4b) and analyzed the expression of downstream targets of Hippo in intestinal tissues. We observed that the mRNA levels of *Ctgf, Cyr61*, and *Ankrd1* were obviously decreased in *Villin-Hhex*[f/f] mice compared with their WT littermates (Fig. 4c). The intestinal mucosa of *Villin-Hhex*[f/f] mice showed fewer Ki67-positive cells than that of their WT littermates (Fig. 4d). Consistent with the above findings, both the number and diameter of intestinal organoids derived from *Villin-Hhex*[f/f] mice were smaller than those of intestinal organoids derived from their WT littermates (Fig. 4e). For colorectal tumor formation, we used an AOM-DSS-induced CRC model (Fig. 4f). Knockout of *Hhex* significantly reduced the number of colitis-induced tumors in a dose-dependent manner (Fig. 4f). Consistent with the above findings, the expression levels of *Ctgf, Cyr61*, and *Ankrd1*, as well as the number of Ki67-positive cells in the tumors, were decreased in *Villin-Hhex*[f/f] mice compared with their WT littermates (Fig. 4g–i). Collectively, these results indicate that HHEX promotes CRC progression.

## CK2 positively regulates the interaction between TEAD4 and HHEX

Among the candidate TEAD4-interacting proteins identified in HCT-116 cells, we noted the alpha subunit of Casein kinase II (CK2), which has been reported to regulate the activity of HHEX in leukemia[13,14] (Fig. 1a). We then examined the potential interaction of CK2 with TEAD4. The

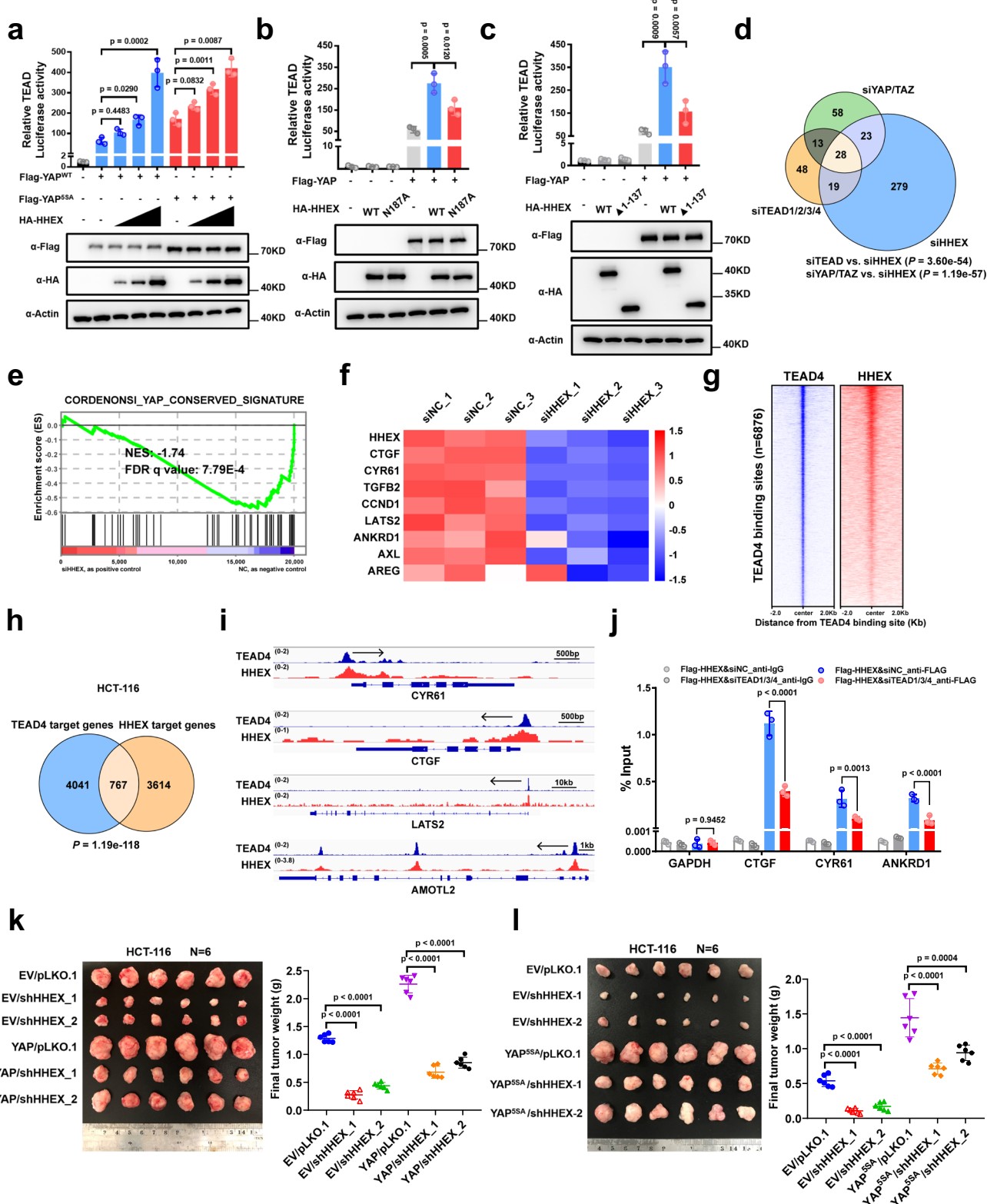

results of the co-IP assay showed that both CK2α and CK2β interacted with TEAD4, which prompted us to explore the potential effect of CK2 on HHEX/TEAD interactions (Figure S6a). Co-IP with cytoplasmic/nuclear fraction further showed TEAD4 mainly interact with HHEX and CK2α/β in nucleus (Figure S6b). Intriguingly, overexpression of CK2α + β but not the kinase-dead CK2α$^{K68M}$ + β dramatically enhanced the interaction of HHEX with TEAD4 (Fig. 5a). In contrast, inhibition of CK2 activity by siRNA targeting *CK2* or the chemical compound

CX-4945, an inhibitor of CK2, attenuated the interaction between HHEX and TEAD4 (Fig. 5b).

Previous studies have shown that CK2 can phosphorylate HHEX at S163/S177, and the phosphomimetic HHEX S163/177E mutant showed reduced nuclear localization in K562 cells[14]. We confirmed and observed that inhibition of CK2 kinase activity by CX-4945 treatment reduced the pan-phosphorylation level of WT HHEX but not the S163/177E mutant (Figure S6c). However, in contrast to the observations in

**Fig. 2 | HHEX promotes the transcriptional activity of the YAP/TEAD complex for tumorigenesis. a** Overexpression of HHEX increased TEAD luciferase activity. Representative immunoblots of the indicated proteins in total lysates were shown. **b, c** TEAD luciferase assay of HA-HHEX WT, N187A mutant (**b**) and N-terminal domain truncation mutant (**c**) in HEK-293T cells. Representative immunoblots of the indicated proteins in total lysates were shown. **d** Venn diagram showing the overlapping downregulated genes among the three gene expression profiles. A hypergeometric test was performed to calculate the statistical significance. **e** GSEA results showing significant enrichment of the YAP target gene signature in *HHEX* knockdown HCT-116 cells. **f** Heatmap showing the mRNA levels of YAP target genes in *HHEX* knockdown HCT-116 cells, as detected by qPCR. $n = 3$ biologically independent samples for the control siNC group. Z-scores of each sample were calculated and shown as the heatmap. **g** Heatmap of ChIP-seq data representing TEAD4 and HHEX binding sites in HCT-116 cells. The heatmap was sorted from the strongest to weakest signal based on TEAD4 binding. **h** Venn diagram showing the overlapping genes with genomic occupancy of both TEAD4 and HHEX in HCT-116 cells. A hypergeometric test was performed to calculate the statistical significance. **i** Representative sequencing tracks of the ChIP-seq data at the *CTGF/CYR61/LATS2/AMOTL2* loci in HCT-116. **j** ChIP-qPCR analysis of FLAG-HHEX binding to the TEAD binding sites in the *CTGF, CYR61,* and *ANKRD1* gene loci in control and *TEAD1/3/4* knockdown HCT-116 cells. The nuclear fractions from ChIP process were used for immunoblots of the FLAG-HHEX and TEAD1/4 (Figure S4j). The *GAPDH* locus was used as the negative control. **k, l** Representative images of xenograft tumors derived from $1 \times 10^6$ HCT-116 cells stably expressing the indicated constructs are shown (left), and the xenograft weights were measured for statistical quantification (right). Data were presented as mean ± SD in this figure. One-way ANOVA with Dunnett's multiple comparison test was performed to assess statistical significance in **a–c, j–l**. $n = 3$ (**a–c, j**), $n = 6$ (**k, l**) biologically independent samples per group. Source data are provided as a Source data file.

K562 cells, both the S163/177C and S163/177E mutants were mainly localized in the nucleus in HCT-116 cells (Figure S6d)[13]. In addition, the phosphomimetic S163/177E mutant exhibited an enhanced interaction with TEAD4, whereas the interaction of the phosphodeficient S163/177 C mutant with TEAD4 was attenuated (Fig. 5c). Interestingly, mutation of S163/177 did not affect the interaction between YAP and HHEX, which further supported that HHEX interacts with YAP and TEAD4 through distinct domains (Figure S6e). Given that serum is a known stimulator of CK2 activity[28,29], we also examined whether the association of HHEX with TEAD4 is responsive to serum stimulation. Indeed, we observed that serum stimulation clearly promoted the formation of the HHEX/TEAD4 complex which was abolished by CX-4945 treatment (Fig. 5d). These results indicate that HHEX forms a complex with TEAD4 in response to phosphorylation by CK2.

Next, we evaluated the functional effect of CK2 on the transcriptional activity of YAP-TEAD complex and the expression of Hippo target genes. Either knockdown of *CK2* or pharmacological inhibition of CK2 with CX-4945 significantly decreased both the enhanced TEAD reporter activity induced by overexpressed HHEX and the mRNA levels of *CTGF, CYR61*, and *AXL* in HCT-116 cells (Figs. 5e, S7a, S7b). Interestingly, we also observed knockdown of *CK2* decreased the mRNA and protein levels of both *YAP* and *TAZ*, but did not affect the expressions of *TEAD4* and *HHEX* in HCT-116 cells, which implicated the potential role of CK2 in transcriptional regulation of YAP/TAZ (Figure S7c). ChIP-qPCR analysis also showed decreased occupancy of HHEX on the TEAD4-bound genomic regulatory regions of *CTGF, CYR61*, and *ANKRD1* upon knockdown of *CK2* (Figs. 5f, S7d). Consistent with these results, co-expression of the HHEX S163/177E mutant with YAP showed enhanced TEAD luciferase activity compared with the WT HHEX and the HHEX S163/177C mutant still activated the TEAD reporter but in less degree (Fig. 5g). To assess whether downregulated expression of YAP/TEAD target genes could be rescued by re-expression of HHEX and its mutants, we took advantage of the third siRNA of *HHEX* which targets the 3'-UTR sequence of the *HHEX* mRNA (Figure S7e). As expected, rescue expression of both WT HHEX and S163/177E mutant restored the downregulated mRNA levels of *CTGF, CYR61*, and *AXL* induced by *HHEX* knockdown in HCT-116 cells, but the S163/177C mutant failed to do so (Fig. 5h). Compared with WT HHEX, the S163/177E mutant showed enhanced activation of Hippo target genes' expression in both control and *HHEX* knockdown HCT-116 cells (Fig. 5h). Intriguingly, overexpression of S163/177C mutant mildly suppressed gene expression of *CTGF, CYR61*, and *AXL* in the control HCT-116 cells (Fig. 5h). This implicates that the S163/177C mutant may exert as a dominant-negative mutant which attenuates the complex formation of endogenous HHEX with YAP/TEAD. Furthermore, the HHEX S163/177E mutant showed enhanced binding but the S163/177C mutant showed attenuated binding to TEAD4-bound genomic regulatory regions (Figs. 5i, S7f). Last, we generated the stable HCT-116 cells expressing S163/177E and S163/177C mutants and tested the pro-tumorigenic effect of these phosphomutants in xenograft assay (Figure S7g). Consistently, the xenografts expressing S163/177E mutant significantly grown larger than the xenografts expressing WT HHEX, while overexpression of the S163/177C mutant suppressed the tumor growth of the HCT-116 xenograft (Fig. 5j). Together, these results indicate that CK2 activates YAP at least partially by phosphorylating HHEX, which in turn binds to and stabilizes the YAP-TEAD complex (Fig. 5k).

## CX-4945 and verteporfin synergistically suppress CRC growth

Our finding that HHEX interacts with both YAP and TEAD to stabilize and activate the YAP-TEAD complex led us to hypothesize that simultaneous disruption of the HHEX-TEAD and YAP-TEAD interactions may result in a synergistic inhibitory effect on the oncogenic transcriptional activity of TEAD. To test this hypothesis, we treated three CRC cell lines with the CK2 inhibitor CX-4945 in combination with verteporfin, a chemical compound targeting the YAP-TEAD interaction. Intriguingly, CX-4945 and verteporfin showed an outstanding synergistic effect in CRC cells, as shown by the combination index (CI) values (CI < 1) (Fig. 6a)[30]. The inhibitory effect of combination treatment with CX-4945 and verteporfin on cell proliferation was more dramatic than that of either single-agent treatment in the CCK-8 assay, consistent with the results of the colony formation assay (Fig. 6a, b). Moreover, combination treatment with CX-4945 and verteporfin induced an increase in PARP1 cleavage in three CRC cell lines (Fig. 6c). Similar results were observed by using a new inhibitory peptide (Super-TDU), which mimics the VGLL4 to disrupt the YAP-TEAD interaction and showed antitumor activity in CRC models[31,32] (Figure S8a–c). In addition, we also tested the synergistic effect of CX-4945 and Super-TDU in HCT-116 cells stably expressing WT HHEX and S163/177E mutant. We observed that the synergistic effect of CX-4945 and Super-TDU was abolished in the cells with overexpression of EE mutant but not WT HHEX (Figure S8d–f). These data further indicate that the antitumor synergistic activity of CK2 inhibitor and YAP/TEAD inhibitor is dependent on disassociation of the HHEX/TEAD complex.

Next, we evaluated the synergistic effect of CX-4945 and verteporfin in a xenograft tumor model. The results showed that combination treatment led to significant regression of HCT-116 tumor growth when compared with vehicle or single-agent treatment (Fig. 6d). IHC staining of Ki67 and cleaved PARP1 in tumors further showed that the combination treatment significantly reduced cell proliferation but increased apoptosis (Fig. 6e). To further assess the therapeutic potential of combination treatment with CX-4945 and verteporfin, we established two patient-derived organoids (PDOs). Consistent with the observations in CRC cell lines, single-agent treatment with CX-4945 or verteporfin led to moderate inhibition of PDO growth, while combination treatment strongly suppressed this growth (Fig. 6f). Taken together, these results show that combined targeting of CK2 and YAP-TEAD may elicit a synergistic antitumor effect in CRC.

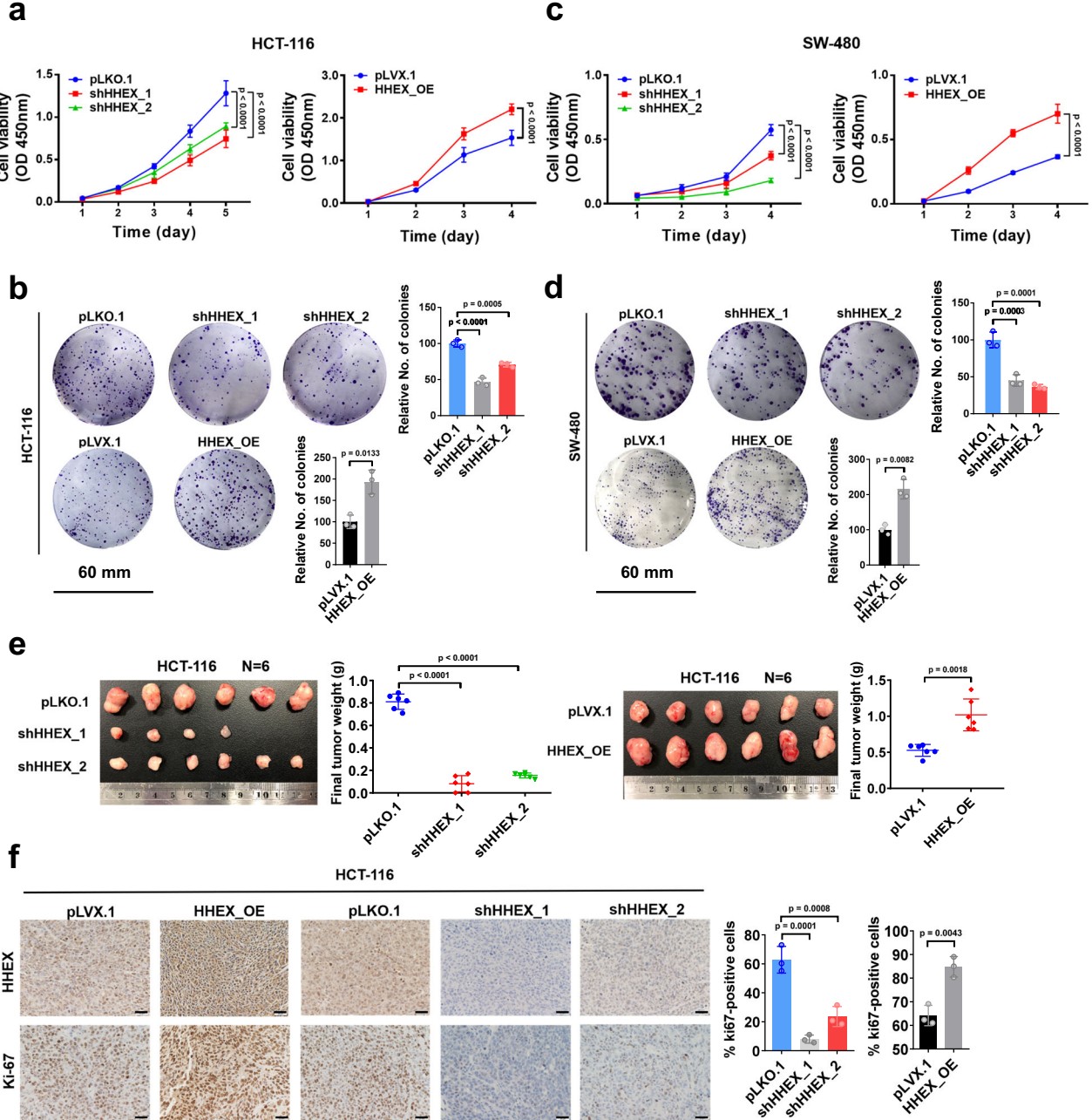

**Fig. 3 | HHEX enhances tumor growth in CRC cells. a, c** CCK-8 assays of the proliferation of HCT-116 (**a**) and SW-480 (**c**) cells with *HHEX* knockdown or overexpression. **b, d** Colony formation assays and statistical analysis of the proliferation of HCT-116 (**b**) and SW-480 (**d**) cells with *HHEX* knockdown or overexpression. Scale bars, 60 mm. **e** Representative images of xenograft tumors derived from $1 \times 10^6$ HCT-116 cells with *HHEX* knockdown or overexpression (*n* = 6). Empty means that no tumors formed. The weights of the xenograft tumors were measured and statistically analyzed. **f** Representative images of IHC staining of HHEX and the proliferation marker Ki67 in xenograft tumors derived from HCT-116 cells with *HHEX*

knockdown or overexpression. The quantification of the number of Ki67-positive cells (%) is shown. Scale bars, 20 μm. Data were presented as mean ± SD in this figure. Data were analyzed by two-way ANOVA with Dunnett's multiple comparison test in **a, c**. One-way ANOVA with Dunnett's multiple comparison test and two-tailed Welch's *t*-test were performed to assess statistical significance for the experiments with >2 groups and 2 groups, respectively, in this figure. *n* = 3 (**b, d, f**), *n* = 6 (**a, c, e**) biologically independent samples per group. Source data are provided as a Source data file.

## HHEX expression is elevated in colorectal cancer and correlates with poor prognosis

Finally, we examined the expression of HHEX in CRC samples and explored its clinical implications. By qPCR analysis of 20 CRC tissues with paired normal mucosal tissues, we observed significantly increased mRNA levels of *HHEX* in CRC (Fig. 7a). Similar results were obtained by western blot analysis of 10 paired primary CRC samples (Fig. 7b). Interestingly, by reanalyzing the gene expression datasets comparing the gene expression profiles between primary CRC and

metastatic CRC (GSE81582, GSE49355, and GSE50760), we found that *HHEX* was differentially expressed in all three datasets (Figure S9a). Next, we confirmed the increased expression of *HHEX* in 6 paired primary and metastatic CRC tissues at both the mRNA and protein levels; these results indicated a potential role of HHEX in tumor metastasis in CRC (Fig. 7c).

To further evaluate the clinical relevance of HHEX in CRC, we performed IHC analysis of CRC tissue arrays consisting of 172 paired CRC and normal tissues. As shown in Fig. 7d, the expression level of

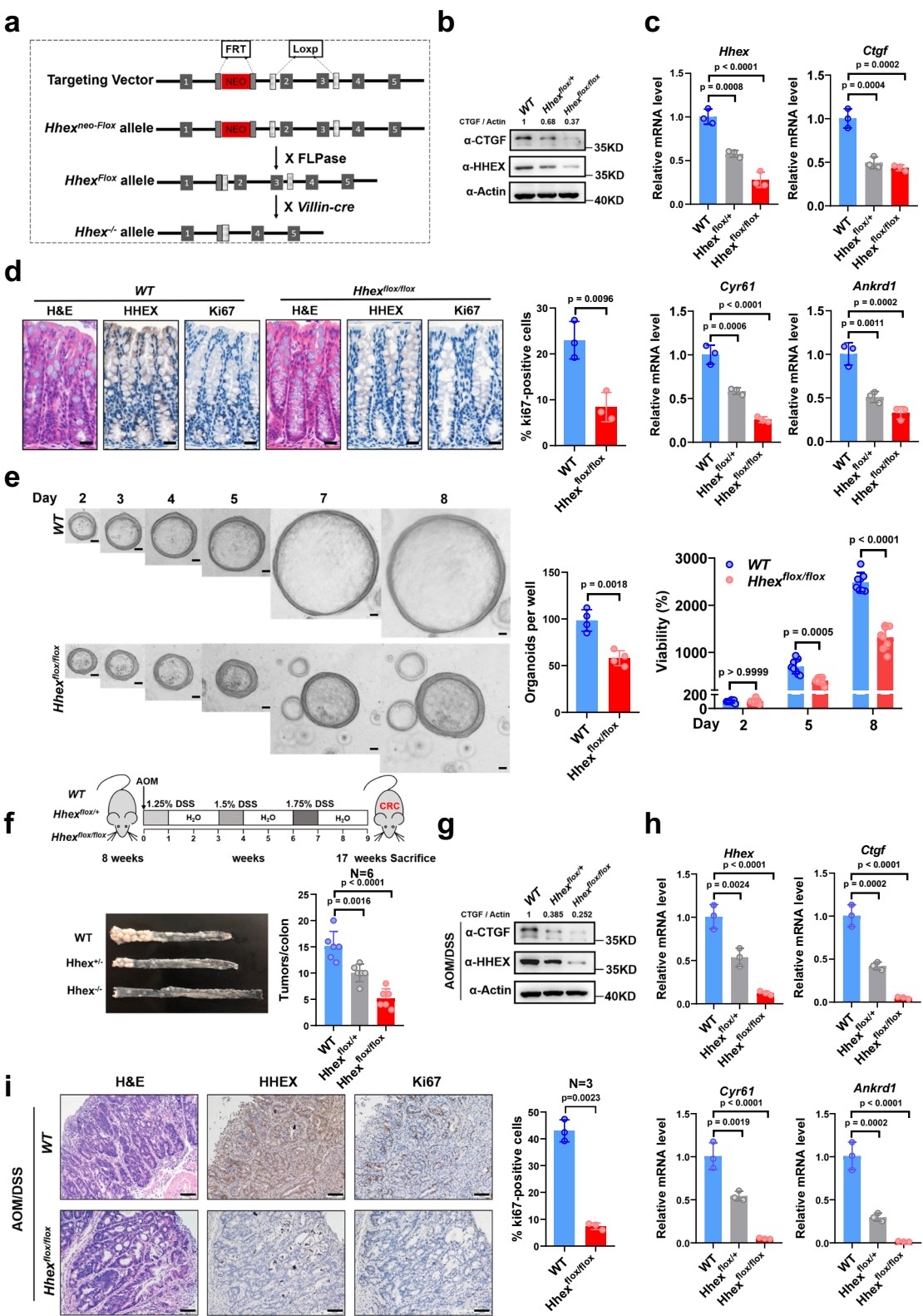

HHEX was relatively low in normal epithelial cells but was significantly increased in tumor cells (Fig. 7d). Furthermore, Kaplan–Meier analysis indicated that a high protein level of HHEX was a marker of poor prognosis in CRC and was associated with shorter overall survival (OS) and disease-free survival (DFS) in CRC patients (Fig. 7e). In addition, we observed that 34.9% of CRC samples had high expression of both HHEX and YAP and that the corresponding patients had the worst OS and DFS

rates (Fig. 7f). Finally, we analyzed the publicly available TCGA dataset and found that the mRNA level of HHEX was positively correlated with the mRNA levels of classical YAP/TEAD target genes, as well as with the YAP signature (a 7-gene signature: *CTGF/CYR61/ANKRD1/AXL/LATS2/TGM2/AMOTL2*) (Figs. 7g, S9b). Taken together, these results indicate that upregulation of HHEX, which is associated with YAP, TAZ, and YAP/TEAD transcriptional activity, is a marker of poor prognosis in CRC.

**Fig. 4 | HHEX knockout reduces CRC tumor formation in vivo. a** Schematic illustration of the generation of *Hhex*^flox/+^ and *Hhex*^flox/flox^ mice. **b** Western blot analysis of HHEX and CTGF protein levels was performed with normal colon tissues harvested from *WT*, *Hhex*^flox/+^, and *Hhex*^flox/flox^ mice. **c** qPCR analysis of *Hhex, Ctgf, Cyr61*, and *Ankrd1* mRNA levels was performed with normal colon tissues harvested from *WT*, *Hhex*^flox/+^, and *Hhex*^flox/flox^ mice. **d** Representative images of H&E staining and IHC staining of HHEX and Ki67 in normal intestinal tissues harvested from WT and *Hhex*^flox/flox^ mice. The quantification of the number of Ki67-positive cells (%) is shown. Scale bar, 50 μm. **e** Images (left) and quantification of the number (middle) and relative viability (right) of organoids derived from WT and *Hhex*^flox/flox^ mice. Black scale bars, 20 μm. **f** An experimental flow chart showing the establishment of the colitis-associated CRC mouse model by AOM/DSS administration. Representative image of tumor-bearing colons from *WT*, *Hhex*^flox/+^, and *Hhex*^flox/flox^ mice.

The colon tumors were counted for statistical analysis. **g** Western blot analysis of HHEX and CTGF protein levels was performed on colon tumor tissues harvested from *WT*, *Hhex*^flox/+^, and *Hhex*^flox/flox^ mice. **h** qPCR analysis of *Hhex, Ctgf, Cyr61*, and *Ankrd1* mRNA levels was performed on colon tumor tissues harvested from *WT*, *Hhex*^flox/+^, and *Hhex*^flox/flox^ mice. **i** Representative images of H&E staining and IHC staining of HHEX and Ki67 in colon tumor tissues harvested from *WT* and *Hhex*^flox/flox^ mice. The quantification of the number of Ki67-positive cells (%) is shown. Scale bar, 50 μm. Data were presented as mean ± SD in this figure. One-way ANOVA with Dunnett's multiple comparison test and two-tailed Welch's *t*-test were performed to assess statistical significance for the experiments with > 2 groups and 2 groups, respectively, in this figure. *n* = 3 (**c, d, h, i**), *n* = 4 (**e**-middle), *n* = 6 (**f**), *n* = 8 (**e**-right) biologically independent samples per group. These data (**b, g**) are representative of 3 independent experiments. Source data are provided as a Source data file.

## Discussion

Hyperactivation of YAP/TAZ and TEAD is commonly observed in CRC[2,17]. The transcriptional activity of the YAP-TEAD complex is normally regulated by the canonical Hippo pathway through modulating YAP/TAZ subcellular localization and protein stability. In addition, multiple transcription factors or transcriptional cofactors, such as AP1, IRF3, and SRF, may cooperate with the YAP/TAZ-TEAD complex[5,6,33]. In this study, we discovered that HHEX, a transcription factor previously known to repress gene expression, acts as a transcriptional coactivator and an agonist of YAP/TEAD in CRC. The homeodomain of HHEX binds to an A/T-rich DNA sequence and represses gene transcription by competing with TATA-box-binding proteins. The homeodomain also mediates the interaction of HHEX with many other transcription factors, such as SRF and AP1[34,35]. Here, we found that the homeodomain of HHEX interacts with TEAD4, while its N-terminal proline-rich domain interacts with YAP. Thus, HHEX stabilizes the YAP-TEAD complex on target gene loci to enhance transcriptional activity. Intriguingly, HHEX mutants defective in binding to either TEAD or YAP still promoted the activity of the YAP-TEAD complex, suggesting that full activation of YAP/TEAD by HHEX requires a dual interaction of HHEX with both YAP and TEAD. It is worth noting that SRF and AP1 have been reported to cooperate with YAP/TEAD to activate gene expression[5,33]. Although HHEX functions as a transcriptional coactivator for SRF in fibroblasts and inhibits Jun-mediated gene activation in teratocarcinoma, it is likely that HHEX coordinates with SRF and AP1 to positively regulate YAP/TEAD activity in CRC.

We found that the DNA-binding TEA domain of TEAD4 is required for its interaction with HHEX and that TEAD4 mutants with abolished DNA-binding ability could not interact with HHEX. However, the interaction between TEAD4 and HHEX is independent of DNA, as digestion with DNase did not disrupt this interaction. The HHEX-SRF interaction has been reported to increase the occupancy of SRF at the promoter of the *SM22α* gene[35]. We speculate that HHEX may similarly enhance the occupancy of TEAD4 at target gene loci. Further studies regarding the biochemical and structural nature of the HHEX-TEAD4-DNA complex are anticipated to shed light on the regulatory mechanism of TEAD-mediated gene transcription and provide structural insights into pharmacological interventions targeting the HHEX-TEAD protein-protein interaction.

The ability of CK2 to phosphorylate the homeodomain of HHEX and therefore inhibit its nuclear localization and DNA binding, eventually inactivating the transcriptional repressor function of HHEX on the VEGF signaling pathway, was first found in K562 leukemia cells[13,14]. The effect of CK2-mediated phosphorylation on the intracellular localization of HHEX might be dependent on the cellular context. In epithelial cells, we did not observe altered intracellular localization of the phosphomimetic HHEX mutant. However, CK2-mediated phosphorylation of HHEX promoted the interaction of HHEX with TEAD4. Furthermore, serum is a strong agonist of YAP/TEAD activity[36]. In addition, CK2 is activated in response to serum and mediates signaling

downstream of serum stimulation[28,29]. In this regard, our current study indicated that serum can activate YAP/TEAD via a CK2-HHEX axis independent of the canonical Hippo pathway. It is worth noting that serum leads to a rapid decrease in YAP phosphorylation in minutes which can be recovered after several hours due to the negative feedback loop of Hippo pathway. In contrast, the effect of serum on interaction between HHEX and TEAD4 occurs after the change of YAP phosphorylation and can last longer time which may account for the long-term maintenance of YAP/TEAD activity induced by serum. Interestingly, in Drosophila, CK2 has been shown to suppress the YAP homolog, Yorkie, and the Yorkie activation-induced overgrowth phenotype[37]. This discrepancy could be due to the possibility that the CK2-HHEX regulatory axis is not conserved in Drosophila or the possibility that CK2 performs different functions in different tissues. In mammals, CK2 is crucial for cancer cell survival and has emerged as a therapeutic target[38]. Our discovery of the CK2-HHEX-TEAD4 regulatory axis is in agreement with the pro-tumorigenic function of CK2 in human cancers. Indeed, we found that treatment with a CK2 inhibitor attenuated the formation of the HHEX/TEAD4 complex and decreased YAP/TEAD activity in CRC cells. The SRC inhibitor dasatinib inhibits leukemic cell survival by decreasing HHEX phosphorylation through indirectly suppressing CK2[11]. Dasatinib is well known as a YAP inhibitor that suppresses SRC-YAP signaling[39]. Our study implies that the inhibitory effect of dasatinib may also rely on an indirect inhibitory effect on the CK2-HHEX-TEAD4 axis in CRC.

Given the overexpression of YAP/TAZ/TEAD and hyperactivation of YAP/TEAD in multiple cancers, inhibition of YAP/TEAD activity constitutes a promising therapeutic strategy for cancer therapy[40]. However, accumulating studies have shown the activation of YAP/TEAD during the administration of chemotherapy, radiation therapy, and targeted therapies for various cancers[40]. The binding interface between YAP and TEAD has been considered as a drug target[40]. Here, we revealed the therapeutic potential of disrupting the interaction between HHEX and TEAD in CRC. Since CK2-mediated phosphorylation of HHEX promotes the formation of the HHEX/TEAD4 complex, inhibition of CK2 constitutes an indirect strategy to disrupt the HHEX/TEAD4 complex and inhibit YAP/TEAD activity (Fig. 8). Importantly, the CK2 inhibitor and YAP/TEAD inhibitor synergistically suppressed CRC growth, indicating that full activation of TEAD requires not only YAP but also other cofactors, such as HHEX.

*HHEX* is a susceptibility gene in T2DM (type 2 diabetes mellitus), and diabetes is known as a risk factor for various cancers, including CRC[41]. Two *HHEX* SNPs that are associated with T2DM have been reported to be associated with increased CRC risk in Chinese patients[41]. Furthermore, HHEX can directly bind to SOX13 to relieve SOX13-dependent repression of Wnt/TCF activity in mouse embryos, indicating a potential oncogenic function of HHEX in CRC[42]. Indeed, our study demonstrated a pro-tumorigenic role of HHEX in CRC. In contrast to the downregulation and aberrant nuclear localization of HHEX identified in breast, prostate, and liver cancers, we found that

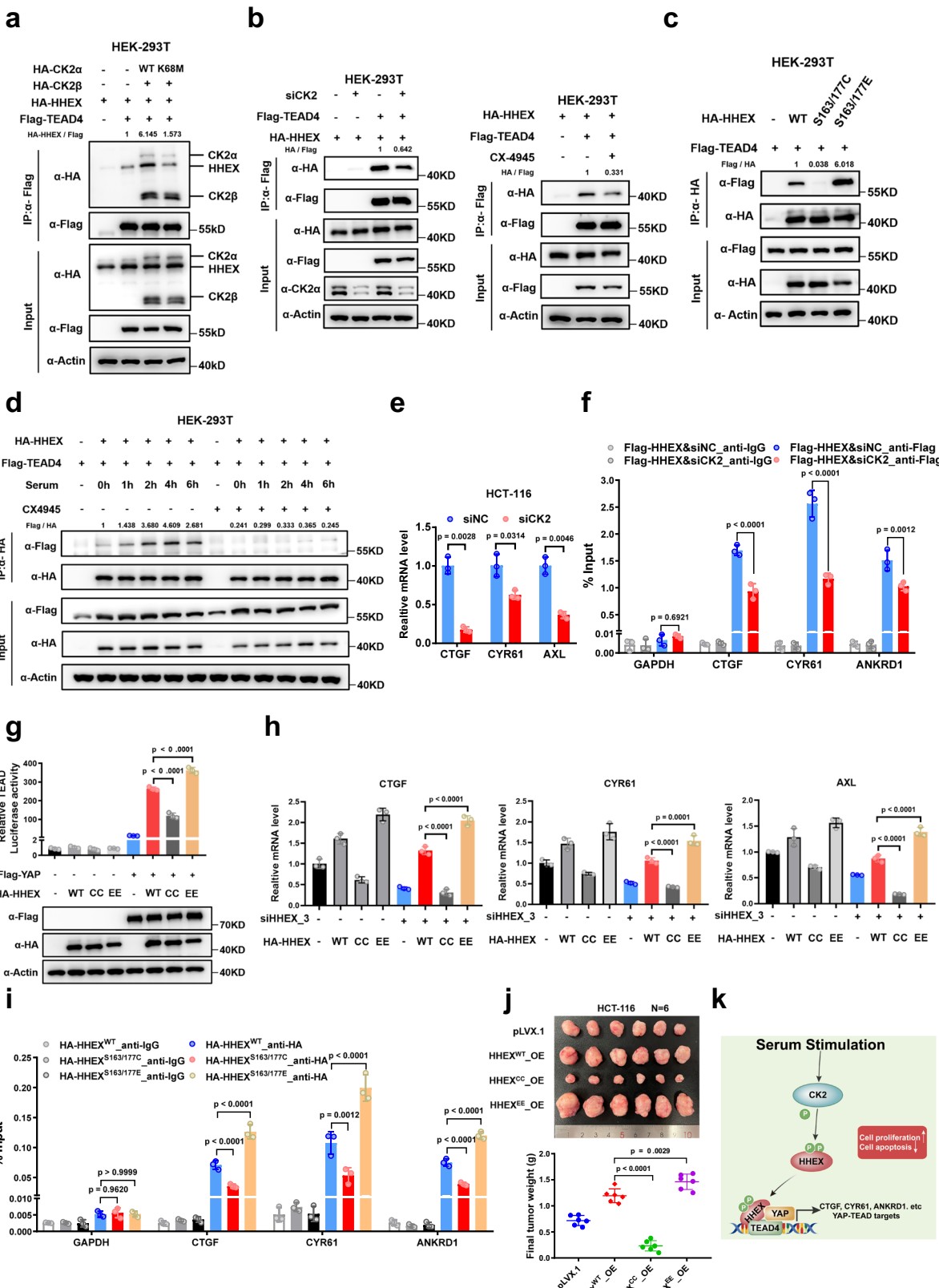

HHEX is upregulated during the progression of CRC and that an elevated expression level of HHEX is correlated with poor prognosis in CRC patients. Therefore, the function of HHEX may depend on the tissue context of specific cancers, which is associated with distinct epigenetic states and gene expression profiles. For example, HHEX inhibits cell proliferation through transcriptional repression of the

VEGF signaling pathway components (*VEGFA, VEGFR1, and VEGFR2*) in leukemic K562 cells, breast cancer MCF7 cells, and HUVECs[10,11,13]. In breast and prostate cancers, HHEX inhibits cell migration through direct transcriptional activation of the TGF-β repressor gene *Endoglin*. In addition, HHEX is considered to play a tumor-suppressive role by repressing *Goosecoid (GSC)* and *ESM1* in several cancers[8,9,12]. However,

**Fig. 5 | CK2 facilitates the interaction between TEAD4 and HHEX.**
**a** Overexpression of the CK2β and CK2α wild type but not the kinase-dead CK2α promoted the interaction between HHEX and TEAD4 in HEK-293T cells. **b** Both knockdown of the *CK2α* and *β* subunits and treatment with the CK2 inhibitor CX-4945 abrogated the interaction between HHEX and TEAD4 in HEK-293T cells. **c** The HHEX phosphomimetic S163/177E mutant exhibited an enhanced interaction with TEAD4 but the phosphodeficient S163/177C mutant exhibited a decreased interaction with TEAD4 in HEK-293T cells. **d** Serum stimulation increased the interaction between HHEX and TEAD4 in HEK-293T cells. Cells were pre-treated with 10 μM CX-4945 for 1 h and then stimulated with 10% serum for indicated time. **e** qPCR analysis of *CTGF, CYR61*, and *AXL* mRNA levels was performed in *CK2* knockdown HCT-116 cells. **f** ChIP-qPCR analysis of HHEX binding to the TEAD binding sites in the *CTGF, CYR61*, and *ANKRD1* genomic loci in control and *CK2* knockdown HCT-116 cells. **g** TEAD luciferase assay of HA-HHEX WT, S163/177C, and S163/177E in HEK-293T

cells. **h** qPCR analysis of *CTGF, CYR61*, and *AXL* mRNA levels was performed in *HHEX* knockdown HCT-116 cells with or without rescued expression of HA-HHEX WT, S163/177C, and S163/177E. **i** ChIP-qPCR analysis of HHEX binding to the TEAD binding sites in the *CTGF, CYR61*, and *ANKRD1* genomic loci in HCT-116 cells expressing WT or mutant HHEX. **j** Representative images of xenograft tumors derived from $1 \times 10^6$ HCT-116 cells with overexpression of the HHEX WT, S163/177C, and S163/177E mutants (*n* = 6). **k** Schematic model showing activation of the YAP-TEAD4-HHEX transcriptional complex by CK2. Data were presented as mean ± SD in this figure. One-way ANOVA with Dunnett's multiple comparison test and Two-tailed Welch's *t*-test were performed to assess statistical significance for the experiments with >2 groups and 2 groups, respectively, in this figure. *n* = 3 (**e**–**i**), *n* = 6 (**j**) biologically independent samples per group. These data (**a**–**d**, **g**) are representative of 3 independent experiments. Source data are provided as a Source data file.

based on RNA-seq analysis of HCT-116 cells, the level of only *VEGFA* was slightly increased upon knockdown of *HHEX* (FC = 1.5), and none of the other HHEX target genes involved in the tumor-suppressive function of HHEX were expressed in CRC cells (GSE196333). In contrast to the observation in cholangiocarcinoma that HHEX can directly activate *NOTCH3* gene expression[16], we found that knockdown of *HHEX* increased the *NOTCH3* level in CRC cells (Supplementary Data 1). These seemingly contradictory roles of HHEX in the regulation of gene expression may be mediated through multiple mechanisms, such as posttranslational modification statuses of HHEX, diverse binding partners of HHEX, and different epigenetic states of target genes in various cancers. The context-dependent differential functions of HHEX warrant further investigation to clarify the complex nature of the interplay between HHEX and other transcription factors. Besides, our study reveals that knockdown of *HHEX* induced downregulated mRNA levels of *YAP/TAZ* but overexpression of HHEX led to upregulation of YAP/TAZ protein levels without change of mRNA levels in CRC cells. Similarly, our study also implicates CK2 could promote gene transcription of *YAP/TAZ* in CRC, which further supports the enhancing effect of CK2 on YAP/TEAD activity and the pro-tumorigenic function of CK2 in CRC. Since HHEX has been reported to bind with eIF4E and inhibit eIF4E-dependent mRNA transport, we hypothesize that HHEX could regulate *YAP/TAZ* at both transcriptional and post-transcriptional level in CRC cells and CK2 might regulate YAP/TAZ through HHEX. Thus, HHEX could also regulate YAP/TEAD activity through indirect mechanism. Furthermore, given the multiple biochemical function of HHEX and the more extensive effect on gene expression profiles by *HHEX* knockdown compared with YAP in CRC cells based on our RNA-seq analysis, we consider that the strong oncogenic effect of HHEX is not entirely dependent on YAP. Thus, further studies are needed to fully elucidate the mechanism of oncogenic function of HHEX and CK2 in CRC in the future. Nevertheless, the coregulatory mechanism of HHEX, YAP, and TEAD4 revealed in this work opens a new possibility of combined targeting of the Hippo pathway for therapeutic purposes.

## Methods
### Cell lines and reagents
All cell lines were purchased from the American Type Culture Collection (ATCC) and validated by STR profiling. All cell lines were cultured in DMEM/high-glucose (HyClone) supplemented with 10% fetal bovine serum, glutamine, and penicillin (Gibco) at 37 °C in 5% $CO_2$. The transient expression and retroviral plasmids of YAP and TAZ have been described in our previous study[43]. The MYC-TEAD1, FLAG-TEAD2, and FLAG-TEAD3 plasmids were kindly provided by Dr. Faxing Yu from Fudan University. The pcDNA3-3XHA-HHEX, pRK7-FLAG-HHEX, pRK7-FLAG-TEAD4, pcDNA3-3XHA-CK2α, and pcDNA3-3XHA-CK2β plasmids were newly constructed for transient transfection by using the ClonExpress II One Step Cloning Kit (Vazyme, China). The HHEX, CK2α, and TEAD4 mutants were generated with a KOD mutagenesis kit

(Toyobo, Osaka, Japan) according to the manufacturer's instructions. Transient transfection was performed by using PEI (Polysciences) or Lipofectamine RNAi MAX (Invitrogen) according to the manufacturer's instructions. For establishing the stable cells overexpressing HHEX, the *HHEX* cDNA was first cloned into pLVX-puro lentiviral vector. In addition, FLAG-HHEX WT, S163/177C, and S163/177E mutants were amplified by PCR with the 5′-primers containing the FLAG-tag sequence, then cloned into pLVX-puro lentiviral vector. *shHHEX-1, shHHEX-2* were generated by using the pLKO.1 vector. The primers and shRNA sequences used in this study are listed in Supplementary Table 1. The retrovirus and lentivirus were generated by transient transfecting the 293T cells with the indicated retrovirus and lentivirus plasmids and related packaging vectors. Virus supernatant was collected twice at 30 and 54 h after transfection. Cells were infected with the virus supernatant in the presence of 8 μg/ml polybrene for 24 h. Then stable cells were selected with treatment with 2 μg/ml puromycin for 1 week or 200 μg/ml hygromycin for 2 weeks before subsequential assays.

CX-4945, Super-TDU, and Verteporfin were purchased from Selleck Chemicals (Houston, TX). The following antibodies were used in this study: HA (3724, Cell Signaling Technology, dilution 1:1000 for WB, 1:300 for IF and 1:100 for ChIP), Flag (14793, Cell Signaling Technology, dilution 1:1000 for WB, 1:300 for IF and 1:100 for ChIP), Myc (2276, Cell Signaling Technology, dilution 1:1000), HHEX (ab34222, Abcam, dilution 1:1000 for WB and 1:100 for IF, PLA, ChIP and IHC), YAP (sc-376830, Santa Cruz Biotechnology, dilution 1:1000 for WB and 1:100 for IP, IF and PLA), p-YAP$^{S127}$ (13008, Cell Signaling Technology, dilution 1:1000 for WB), TAZ (560235, BD, dilution 1:1000 for WB and 1:100 for IF), TEAD4 (ab58310, Abcam, dilution 1:1000 for WB and 1:100 for PLA), TEAD1 (A6768, Abclonal, dilution 1:1000 for WB), Pan-TEAD (13295, Cell Signaling Technology, dilution 1:1000 for WB, 1:100 for IP), CTGF (A11067, Abclonal, dilution 1:1000 for WB), Ki67 (D3B5,Cell Signaling Technology, dilution 1:100 for IHC), CK2α (10992-1-AP, Proteintech, dilution 1:1000 for WB), Cleaved parp-1 (ab32064, Abcam, dilution 1:1000 for WB and 1:100 for IHC), β-actin (A2228, Sigma-Aldrich, dilution 1:10,000 for WB), β-Tubulin (2128, Cell Signaling Technology, dilution 1:1000 for WB), LaminA/C (4777, Cell Signaling Technology, dilution 1:1000 for WB), Normal Rabbit IgG (2729, Cell Signaling Technology, for IF and ChIP), Rabbit mAb IgG Isotype Control (3900, Cell Signaling Technology, for IP, PLA and ChIP), Mouse mAb IgG Isotype Control (5415, Cell Signaling Technology, for IP, IF, and PLA).

### Human tissue samples and immunohistochemistry
All human CRC samples were collected between August 2008 and November 2018 at Xinhua Hospital, Shanghai Jiao Tong University School of Medicine. Postsurgical follow-up was conducted until August 2020. Institutional review board approval and informed consent were obtained for all sample collections. The CRC tissue array was customized constructed by the TOPGEN company, Shanghai[44]. IHC staining of the samples was performed according to the general

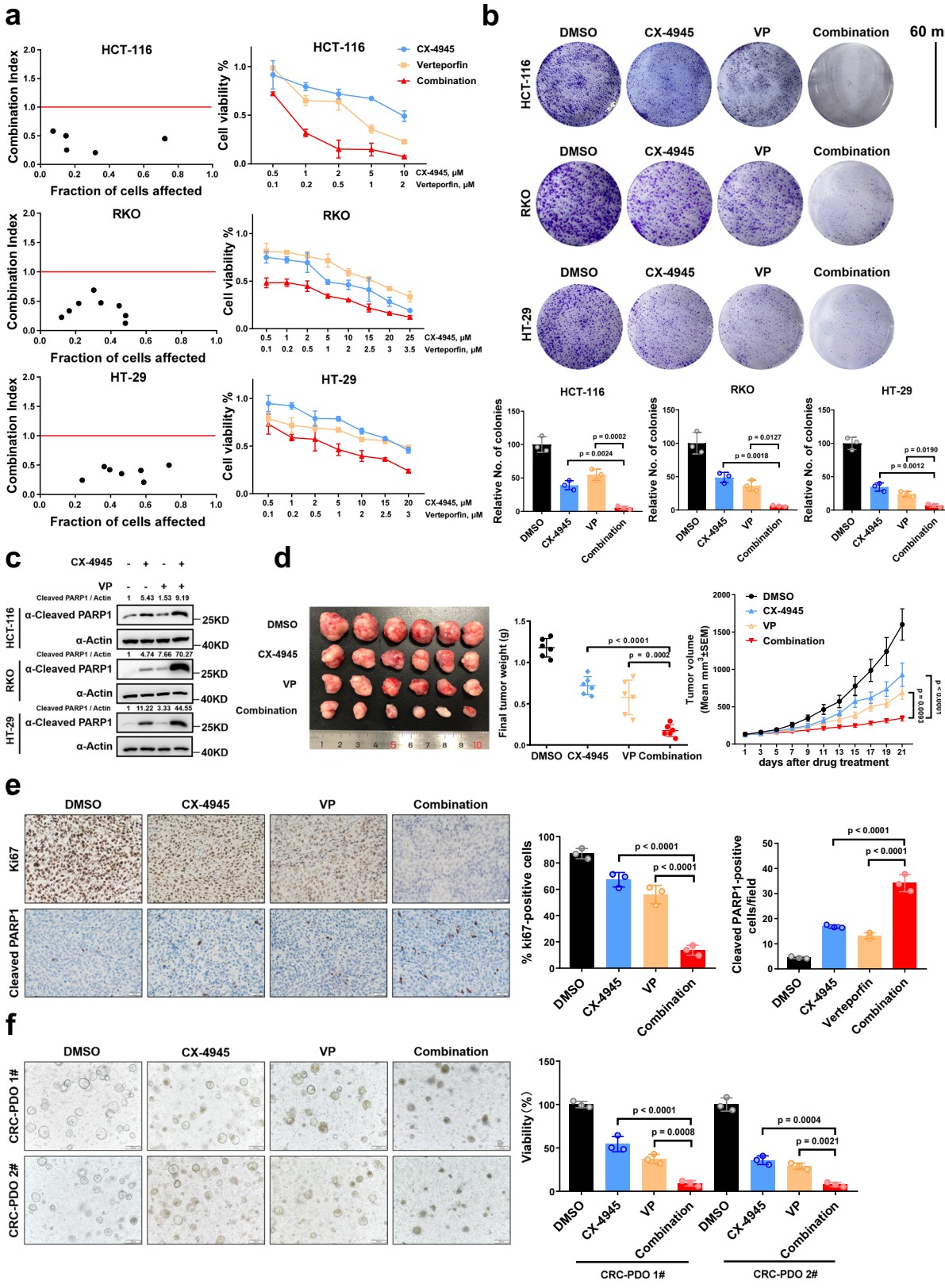

protocol by using the heat-induced epitope retrieval method. The semiquantitative analysis was performed as followed: The staining intensity was scored as 0 (lack of staining), 1 (weak staining), 2 (moderate staining), or 3 (strong staining), and the percentage of staining was scored as 1 (<25%), 2 (25−50%), 3 (50−75%), or 4 (>75%). IHC staining was assessed by two independent pathologists. Low and high expression of HHEX were defined as IRS < 6 and ≥ 6, respectively.

**Proximity ligation assay**

A proximity ligation assay (PLA) was performed using a Duolink In Situ Detection Kit (Sigma, #DUO92008) according to the manufacturer's instructions. Briefly, cells plated on glass coverslips were fixed, permeabilized, blocked, and incubated with the indicated primary antibodies. Then, cells were hybridized to PLA probes. After ligation and amplification of PLA signals, the PLA puncta were photographed.

**Fig. 6 | Synergistic antitumor activity of CX-4945 and verteporfin in CRC. a** Cell viability was assessed after 48 h of exposure to the indicated concentrations of CX-4945 and VP alone or in combination in HCT-116, RKO, and HT-29 cells. CCK8 was used to detect the cell viability. CI (combination index) values were calculated using CompuSyn. **b** Representative images of the colony formation assay. HCT-116, RKO, and HT-29 cells were treated with the CK2 inhibitor CX-4945 (10 µM) or the YAP/TEAD inhibitor VP (2 µM) alone or in combination. Scale bars, 60 mm. **c** Western blot analysis of cleaved PARP1 in HCT-116, RKO, and HT-29 cells treated with CX-4945 (10 µM) or VP (2 µM) alone or in combination for 48 h. **d** Representative images of HCT-116 xenograft tumors excised from nude mice in the different groups (left). Tumor weights in the different groups of mice (middle). Tumor growth curves of HCT-116 CDXs (right). Two-way ANOVA was performed to assess statistical significance of the tumor growth curves. **e** Representative images of IHC staining of Ki67 and cleaved PARP1 in HCT-116 xenograft tumors treated with vehicle or with CX-4945 or VP alone or in combination. The quantification of the number of Ki67-positive cells (%) and cleaved PARP1 are shown. Scale bars, 20 µm. **f** Representative photomicrographs of two CRC PDOs (PDO #1-2) treated with CX-4945 (10 µM) or VP (2 µM) alone or in combination for 48 h. Scale bars, 100 µm. Cell viability was measured by a CellTiter-Glo assay. Data were presented as mean ± SD in this figure. One-way ANOVA with Dunnett's multiple comparison test was performed to assess statistical significance in this figure. $n = 3$ (**b, e, f**), $n = 4$ (**a**), $n = 6$ (**d**) biologically independent samples per group. Source data are provided as a Source data file.

## Immunofluorescence staining assay (IFA)

Cells were seeded on the glass coverslips one night before, and then fixed in 4% paraformaldehyde. After permeabilization with 0.5% Triton X-100 at room temperature for 10 min, cells were blocked by 5% BSA for 1 h, and then incubated with primary antibody overnight. Subsequently, FITC-conjugated secondary antibody (Invitrogen A11008, A11012, A11005) was applied for 1 h away from light and nucleus was stained by DAPI for 30 min. Immunofluorescence was visualized by Olympus IX81.

## Luciferase reporter assay

Cells were plated on 24-well plates and transiently co-transfected with TEAD luciferase reporter (Addgene, 34615) and indicated plasmids for 24 h. For the treatment of CK2 inhibitor CX-4945, cells were transfected with indicated plasmids for 6 h and then incubated with CX-4945 for the next 18 h. For the knockdown of *CK2*, cells were co-transfected with CK2 siRNA and indicated plasmids for 48 h. The luciferase activity was measured by using a dual luciferase reporter assay (Promega) and normalized to the activity of Renilla luciferase.

## Western blot analysis and immunoprecipitation

For direct western blot analysis, cells were harvested in NP-40 lysis buffer (1% NP-40, 50 mM Tris-HCl at pH 7.5, 150 mM NaCl, 1 mM PMSF, 25 mM NaF, 1 mM Na₃VO₄) supplemented with cOmplete™ Protease Inhibitor Cocktail (Roche). For immunoprecipitation, cells were harvested in 0.3% NP-40 lysis buffer and the cell lysate was incubated with anti-FLAG/HA/MYC magnetic beads (Bimake) for 3 h at 4 °C. For the DNase treatment experiment, the cell lysate was first treated with Benzonase (Sigma) (with 2 mM MgCl₂) for 0.5 h at 37 °C and immunoprecipitation was performed as above description. For the endogenous immunoprecipitation, the cell lysate was incubated with indicated antibodies and control IgG for 1 h and protein A/G agarose for another 2 h at 4 °C. The precipitated protein was eluted from beads with 80 µl 1X loading buffer after boiling 10 min at 95 °C.

## RNA sequencing, qPCR

Total RNA from HCT-116 cells transfected with NC siRNA (5′-UUCUCCGAACGUGUCACGUTT-3′), *siYAP-1 + siTAZ-1* (*siYAP-1*: 5′-GACAUCUUCUGGUCAGAGA-3′, *siTAZ-1*: 5′-ACGUUGACUUAGGAA-CUUU-3′), *siYAP-2 + siTAZ-2* (*siYAP-2*: 5′-CUGGUCAGAGAUACUUCUU-3′, *siTAZ-2*: 5′-AGGUACUUCCUCAAUCACA-3′), *siTEAD1/2/3/4* (*siTEAD1/3/4*: 5′-UGAUCAACUUCAUCCACAA-3′, *siTEAD2*: 5′-GCCAGAUGCA-GUUGAUUCUTT-3′), *siHHEX-1* (5′-GUGAUCAGAGGCAAGAUUUTT-3′), *siHHEX-2* (5′-GGAUAGCUCUCAAUGUUCGTT-3′), *siHHEX-3* (5′-CCCA-CUUAAUGGAAAGGCAAA-3′) was extracted and used for RNA-seq on the HiSeq 2500 platform. Differentially expressed genes were analyzed by DESeq software and confirmed by qPCR. The raw data were deposited in the Gene Expression Omnibus (GSE196333). shRNA sequence of *HHEX* and other siRNA sequence are listed in Supplementary Table 1. All primers are listed in Supplementary Table 2.

## ChIP and ChIP-seq analysis

HCT-116 cells were plated in low density ($1 \times 10^6$ cells/10 cm plate) and 10 plates of HCT-116 cells were then harvested for Chromatin immunoprecipitation (ChIP) by using the Magna ChIP kit (Merk, 17-610). Briefly, cells were cross-linked with 1% formaldehyde for 10 min at room temperature. The reaction was stopped by adding 1.25 M glycine for 5 min at room temperature. After washing, nuclear fraction, nuclear lysis, and sonication, the chromatin was sheared into 100–500 bp fragments. And then, the chromatin fraction was incubated with anti-HHEX (ab34222, Abcam), anti-FLAG (14793, Cell Signaling Technology), Rabbit mAb IgG Isotype Control (3900, Cell Signaling Technology) or Normal Rabbit IgG (2729, Cell Signaling Technology) overnight at 4 °C. Chromatin-bound beads were subjected to extensive washing and the eluted chromatin was de-cross-linked with ChIP elute buffer with proteinase K for 2 h at 65 °C. The eluted DNA was purified for subsequent qPCR analysis. The qPCR primers of control and YAP target genes were included in Supplementary Table 2. For the HHEX ChIP-seq analysis, the ChIP enriched DNA and Input DNA samples were prepared to generate libraries and sequenced by the Illumina HiSeq 2500 platform. The raw data were deposited in the Gene Expression Omnibus (GSE196333). After quality control, the clean reads were aligned to the human reference genome (GRCh38) by the Bowtie 2[45]. MACS2 was used to call peaks with Input sample as the negative control[46]. Due to there is no duplicate samples for HHEX ChIP assay, *q*-value < 0.05 was used as threshold to call peaks. The ChIP-seq datasets of TEAD4 in HCT-116 (ENCSR000BVJ) and HepG2 (ENCSR000BRP) and HHEX in HepG2 (ENCSR656JZL) were downloaded from ENCODE database. The irreproducible discovery rate (IDR) threshold of 0.02 was used to assess the consistency of replicate experiments and to obtain a high-confidence single set of peak calls for ENCODE datasets according to the guideline of ENCODE[47]. Heatmaps were generated by DeepTools2 that considers a 2 kb window centered on peak summits[48]. Integrated Genomics Viewer (IGV) was used to visualize ChIP-seq profiles[49].

## Cell proliferation and colony formation assays

Cell proliferation was measured by Cell Counting Kit-8 (CCK8) according to the manufacture's protocol. Stable cells were seeded at a density of 1000 cells per well in 96-well plates, and the cell viability was measured by Cell Counting Kit-8 every day. For drug treatment, CRC cells were plated in 96-well plates at a density of 3000–5000 cells per well. After cell attachment for one day, CRC cells were treated with indicated drugs at various concentrations for 48 h. Then, the CCK8 was used to evaluate the cell viability. The culture medium with corresponding concentration of indicated drugs was also measured by CCK8 and used as the background control. Combination effects were analyzed by the Chou-Talalay combination index (CI) method using CompuSyn software[30]. The CI values of <1, =1, >1 indicate synergism, additivity, and antagonism between the drugs, respectively. For the colony formation assay, stable cells were seeded at a density of 1000 cells per well in 6-well plates for 2 weeks. For the colony formation assay with drug treatment, CRC cells were seeded into 6-well plates (5000–10,000 cells per well) and cultured with the indicated drugs for

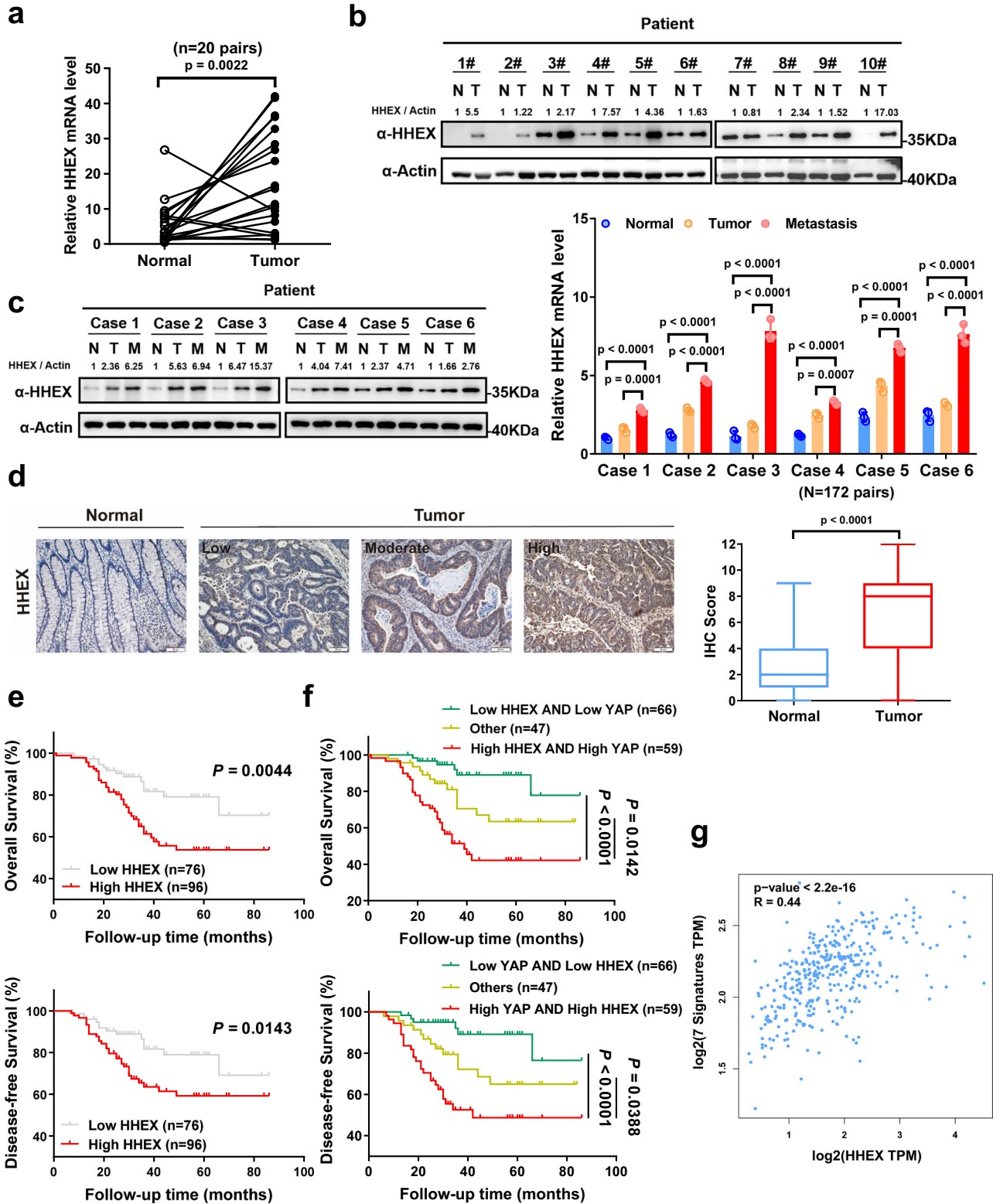

10 days. Then, cells were fixed in 4% paraformaldehyde at room temperature for 30 min and stained with 0.1% crystal violet for 30 min. After washing with PBS, the number of colonies was counted for statistical analysis.

**Mouse xenograft experiment**
All mouse studies were approved, and all animals were manipulated according to the protocols approved by the Animal Care and Use Committees of Xinhua Hospital and animal care was conducted in accordance with institutional guidelines. According to the criteria of the Animal Care and Use Committee of Xinhua Hospital, the maximal tumor burden permitted was <10% body weight, at no point did any mice exceed maximal tumor burden. Mice were housed in pathogen-free and ventilated cages, and allowed free access to irradiated food and autoclaved water ad libitum in a 12 h light/dark cycle, with room temperature at 21 ± 2 °C and humidity between 45 and 65%. Male BALB/

**Fig. 7 | The clinical implications of HHEX expression in CRC. a** The mRNA expression levels of *HHEX* in 20 pairs of CRC and adjacent normal tissues were determined by qPCR analysis. Paired Student's *t*-test was performed to assess statistical significance. **b** Western blot analysis of HHEX protein levels in 10 pairs of CRC and paired adjacent normal tissues. The data are representative of 3 independent experiments. **c** Western blot analysis (left) and qPCR analysis (right) of *HHEX* expression levels in six sets of matched hepatic metastases, primary tumors and normal tissues. *n* = 3 technical triplicate per group. Data were presented as mean ± SD. One-way ANOVA with Dunnett's multiple comparison test was performed to assess statistical significance. **d** Representative images (left) and statistical analysis (right) of IHC staining of HHEX in 172 pairs of normal and CRC tissues (*n* = 172 biologically independent CRC samples). Scale bars, 50 μm. The whiskers of boxplot represent the quantile percentile, from bottom to top are minima, 25%,

median, 75%, and maxima respectively. Wilcoxon signed-rank test was performed to assess statistical significance. **e** Kaplan–Meier plots of the overall survival and disease-free survival of CRC patients stratified by the HHEX protein level. *p* = 0.0044 (OS) and *p* = 0.0143 (DFS) by two-sided the log-rank test. **f** Kaplan–Meier plots of the overall survival and disease-free survival of CRC patients stratified by the protein levels of HHEX and YAP. *p* = 0.0142 (OS, HHEX high/YAP high vs other), *p* < 0.0001 (OS, HHEX high/YAP high vs HHEX low/YAP low), *p* = 0.0388(DFS, HHEX high/YAP high vs other) and *p* < 0.0001 (DFS, HHEX high/YAP high vs HHEX low/ YAP low) by two-sided the log-rank test without adjustment. **g** Positive correlation between the mRNA level of *HHEX* and the YAP target gene signature in colorectal cancer (*n* = 367 biologically independent CRC samples). Analysis was performed by the GEPIA2 database. *p* < 2.2e−16 by two-sided Pearson correlation analysis. Source data are provided as a Source data file.

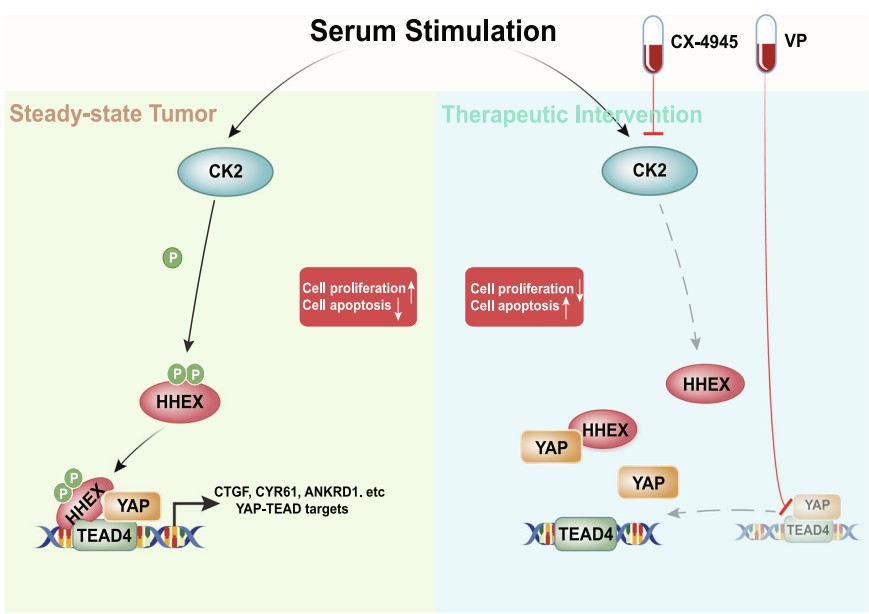

**Fig. 8 | Schematic diagram of the findings in this study.** Model of how HHEX activates YAP/TEAD4-dependent transcription in response to serum stimulation.

c nude mice of ~4–6 weeks of age were purchased from GemPharmatech, Shanghai, China. Mice were subcutaneously injected with $1 \times 10^6$ cells of indicated stable CRC cell lines resuspended in 100 μl phosphate-buffered saline (PBS) into the right flank (*n* = 6 mice per group) and were sacrificed on day 21. For the experiments assessing the xenografts overexpressing YAP^SSA and HHEX, the mice were sacrificed on day 18 to avoid excess tumor burden. For drug treatment, after the volume of xenografts reached approximately 100 mm³, the mice were randomly assigned into the indicated groups and the mice were administrated with DMSO vehicle, CX-4945 (25 mg/kg, oral gavage twice daily), VP (25 mg/kg, intraperitoneal injection every other day) or a drug combination in which each compound was administrated at the same dose and scheduled as single agent for 3 weeks (*n* = 6 mice per group). The tumor volume based on caliper measurements was calculated by the formula: 0.5 × (largest diameter) × (smallest diameter)². At the experimental endpoint, the mice were sacrificed, and the xenografts were dissected, photographed and weighted. Then the xenografts were fixed in 4% paraformaldehyde overnight at room temperature before paraffin embedding and sectioning for IHC analysis.

### Mice and model of AOM/DSS-induced CRC
*Hhex*^flox/flox mice and *Villin*-cre mice which in a C57BL/6 background were purchased from Jackson Laboratories (Stock No: 025396 and 004586, respectively) were crossed to generate *Villin-Hhex*^flox/flox mice

for the AOM/DSS CRC model. Eight-week-old mice were injected intraperitoneally with 10 mg/kg AOM (Sigma) on day 1, followed by three cycles of treatment with DSS (MP Biomedicals) dissolved in drinking water at stepwise increasing concentrations of 1.25%, 1.5%, and 1.75% beginning on day 2. In each DSS cycle, mice drank DSS water for 7 days and then drank regular drinking water for 2 weeks. Mice were sacrificed for harvesting of colon tissues on day 65. The tumors in the colon were measured and photographed. Then, the whole colon was sectioned for IHC analysis or collected for the protein and RNA extraction.

### Three-dimensional (3D) organoid culture and CellTiter-Glo 3D viability assay
Patient-derived organoid (PDO) culture, mouse organoid culture, and organoid viability assays were performed according to the initial reference[50]. The whole intestine from untreated WT and *Villin-Hhex*^flox/flox 4-week-old mice was used to establish mouse organoids. Mouse organoids were first seeded in 24-well plates and cultured for 8 days before counting the number of the organoids (Diameter ≥ 20 μm). For assessing the viability of organoids, mouse organoids were plated in 96-well plates. The viability of organoids was measured by CellTiter-Glo luminescent cell viability assay (Promega, USA). Institutional review board approval and informed consent were obtained for all CRC samples used for establishing the CRC PDOs. PDOs were first established and cultured in 24-well plates, and re-

plated in 96-well plates for drug treatment (*n* = 3 wells/group). After 24 h, the medium was replaced with fresh medium containing DMSO or containing CX-4945 or VP alone or in combination for another 72 h. 3D organoid viability was quantified using CellTiter-Glo (Promega). Relative viability was normalized to that in the DMSO group.

## Statistical analysis
The data were collected by using Gelpro 32 (v4.0), Microsoft Excel 2019, FastQC (v0.11.9), MACS2 (v2.2.7.1), R package DESeq2 (1.26.0), and ImageJ Launcher (v1.4.3). All statistical data were analyzed and plotted with GraphPad Prism 8.0. Quantitative data are presented as the mean ± standard deviation (SD) values. For two group comparisons between two groups, statistical analysis was performed using the two-tailed Welch's *t*-test. One-way ANOVA with Dunnett's multiple comparison test was used to assess the statistical significance for the experiments with >2 independent groups. For the CCK8 and xenografts growth curve assays, two-way ANOVA with Dunnett's multiple comparison test was performed to assess the statistical significance. Paired Student's *t*-test was performed to assess statistical significance of *HHEX* mRNA differential expression in 20 pairs of CRC and adjacent normal tissues. For the paired IHC score data, Wilcoxon signed-rank test was used. Pearson correlation analysis was used to assess statistical significance between the mRNA levels of *HHEX* and seven YAP target genes in TCGA colorectal cancer datasets.

## Reporting summary
Further information on research design is available in the Nature Research Reporting Summary linked to this article.

## Data availability
RNA-seq and ChIP-seq primary data generated in this study have been deposited in the GEO database under accession code GSE196333. And the human reference genome (GRCh38) was used in ChIP-seq analysis. The ChIP-seq datasets of TEAD4 in HCT-116 and HepG2 and HHEX in HepG2 data used in this study are available in the ENCODE database under accession code ENCSR000BVJ, ENCSR000BRP, ENCSR656JZL separately. All the data supporting this study are available within the article, the Supplementary file, the Source data file, as indicated in the Reporting summary for this article. A Reporting summary for this article is available as a Supplementary Information file. Source data are provided with this paper.

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

## Acknowledgements

This work was sponsored by the National Key R&D Program of China (2019YFC1316002), the National Natural Science Foundation of China (81873547, 82172916, 81874177, 81974060, 82073056, and 82073201), Innovative research team of high-level local universities in Shanghai (SSMU-ZDCX20181202), and the Program for Professor of Special Appointment (Young Eastern Scholar) at Shanghai Institutions of Higher Learning (to C.-Y.L.).

## Author contributions

Y.G.G., Z.H.Z., and Z.Y.H. performed experiments and analyzed data; L.C. analyzed clinical patients' data; W.Y. and W.J.H. designed study and analyzed data; Z.C.Z., P.D., and C.Y.L. conceived and supervised the study; YG.G., Z.C.Z., and C.Y.L. wrote the paper.

## Competing interests

The authors declare no competing interests.
