## [Peer Review File · Nature Communications]

CK2-induced cooperation of HHEX with the YAP-TEAD4 complex promotes colorectal tumorigenesisReviewer's Comments:

Reviewer #1 (Remarks to the Author)

Reviewer comments

The article by Guo Y. and Zhu Z. et al investigates the cooperation of HHEX with the YAP-TEAD complex and highlights its role in activation of a YAP dependent gene set in colorectal cancer. The authors suggests that HHEX can co-activate a YAP-TEAD driven transcriptional program in a CRC cell line and therefore HHEX possess oncogenic features in CRC. Multifaceted biochemical experiments regarding binding interactions and domain characterizations were carried out and the authors propose that phosphorylation of HHEX by casein kinase 2 (CK2) enhances its binding to TEAD-YAP. Treatment of the CK2 inhibitor, CX-4945, has synergistic effects in CRC when combined with the YAP-TEAD inhibitor verteporfin. However, the mechanistic experiments supporting this mechanism are not very well worked out and are largely correlative. A detailed list is given below. Given the potential important role of HHEX in CRC tumorigenesis, the reviewer recognizes the contribution to the understanding of YAP-TEAD transcriptional control. However, because of the extensive in vivo and in vitro experiments that need to be performed to support their model and conclusions, the reviewer would not recommend this article for a revised version. In case, the authors will hand in a revised version of this manuscript, please provide a reviewer link for their NGS data.

The authors should address the issues below:

Major points:

General comments:

1. In most of the experiments information about the number of replicates are missing, e.g. figure 2A, B, C, F, G; figure 3A, B, C, D, F; figure 4C, D, E, I (ki67 panel); figure 5 E, F, G; figure 6 A, B, E, F. Bar plots should include individual data points like done for Fig4E/F and number of replicates should be included in the figure legend.
2. In all Co-IP experiments, loading controls are missing! CoIP experiments where authors showed enhanced or reduced binding of HHEX to TEAD should be quantified, e.g. Fig 1E, 5A, 5B, 5C
3. ChIP-Seq is required to demonstrate the general co-occupancy of TEAD/YAP/TAZ and HHEX.

Specific:

4. Fig. 1C: How did the authors control for DNase activity in this assay? The alternative conclusion here would be that the DNase treatment did not work. Given the involvement of DNA binding domains in the interaction, this point is critical.
5. Fig1E: Can the results from the CoIP studies be quantified to better estimate the effect of HHEX expression on the YAP-TEAD interaction? Also, it would be nice to have a loading control in this experiment, to prove equal loading of the gel.
6. Fig. 1F: The authors should remove the model at this position. Here, there could be thousand different ways how HHEX affects TEAD/YAP/TAZ.
7. Fig. 1G: An immunoblot to demonstrate HHEX KD efficiency/ HHEX oe is absolutely required.
8. Fig1H: Quality of the PLA assay is rather low. Pictures look like the assay did not work probably, because there are also foci outside the nucleus and a lot of background. In addition, the quality of the controls used in this assay is not appropriate. As this is the only experiment proving endogenous interaction between HHEX-TEAD and HHEX-YAP, this should be crystal clear. Can the authors provide PLA assay using single antibody controls (either single anti-HHEX or single anti-YAP or single anti TEAD). Preferable would be an PLA assay after knockdown of HHEX to prove that the signal is specific.
9. The authors should perform endogenous HHEX TEAD/YAP/TAZ CoIPs.
10. Fig. 2A-C: Immoblots for the transfected proteins are required. What about the F32E mutant that still interacts with TEAD/Y/T? Does this behave as a stronger activator?
11. Fig.2D: The number of differentially regulated genes for a Y/T KD is minimal indicating

technical issues. That result is very confusing/concerning. How can HHEX Kd have such a strong effect on YAP-dependent tumor growth even though a minimal number of genes are affected? Here, also immunoblots for all samples are absolutely required. Maybe KD of HHEX also affects TEAD Y/T protein levels. A hypergeometric test is required to demonstrate the significant overlap of the DGEs.

12. The reviewer asks the authors for GEO access for the data

13. Fig. 2E: Here, the labels are missing. The way the GSEA is drawn here, it would demonstrate a significant UPREGULATION of YAP target genes. However, according to their model those genes should be downregulated after HHEX knockdown?

14. Fig2F: Apparently, the authors performed qPCR analysis on duplicates? If so, this is not sufficient for meaningful statistical analysis. The heat map should depict Z-scores to account for variance.

15. Fig2G: Please include a negative control region in the experiment, only IgG control is not sufficient. Also, it would be interesting to know whether HHEX binding is not affected by TEAD depletion at HHEX "only" genes. Please provide data for that. Again, immunoblots are missing in order to draw meaningful conclusions.

16. Fig2H/I also Fig3: The effect of shHHEX on tumor growth looks very convincing. However, here again HHEX immunoblots are absolutely required. Furthermore, this experiment does not demonstrate that the effect of HHEX is YAP-dependent, since HHEX on its own has dramatic effects. The authors should repeat the HHEX oe with the S163/177 phosphomutants (dead and mimetic vs. wt).

17. Fig.2/3: Do these tumors really depend on YAP/TAZ for proliferation? Here, also Y/T-independent CRC cell lines are required as ctrl. For this, see the recent Cancer Cell paper by the Bremner lab.

18. Fig. 3F: To me, these pictures do not look very "representative". E.g. there is not a single Ki-67 positive cell in the shHHEX_1 sample but the quantification shows 20%.

19. Fig. 4A: Is there an effect on Y/T or TEAD protein or Y/T phosphor status?

20. Fig5A: The authors show a direct interaction between CK2, HHEX and TEAD. However, in your model (Fig. 5H) CK2 is in the cytoplasm and phosphorylates HHEX. How can it then interact with TEAD4?

21. Fig. 5A: Is the increased interaction dependent on the catalytic activity of CK2?

22. Fig5B: Blot showing CK2 expression is missing in this experiment. Data provided in Sup Fig. 6C is also not convincing since there is almost no difference in the CK2 expression after siCK2 treatment. Also, there is a discrepancy between the knockdown efficiency on protein and RNA level.

23. Fig 5 in general: Effect of the inhibitor CX-4945 as well as siCK2 seem to have only little effects on the HHEX-TEAD interaction (Fig.5B), whereas the interaction is completely abolished when S163/177 are mutated (Fig.5C).

24. Do the HHEX mutants S163/177C and S163/177E, siCK2, CX-4945 treatment affect relative TEAD luciferase activity similar to experiments done in Fig.2?

25. After knockdown of HHEX in HCT-116 cells, can the expression of YAP target genes be restored by WT HHEX in comparison to the CC or EE HHEX mutants? One would expect that CC could not restore and EE is doing a better job than WT.

26. Does knockdown of CK2 affect YAP expression, stability or localization and therefore the complex of YAP-TEAD-HHEX is less stable?

27. Fig. 6: General comment for the use of the inhibitors. The authors do not provide any evidence, that CX inhibition even does what they propose it does, namely that it hits HHEX S163/177. This they have to convincingly demonstrate. Otherwise, Fig. 6 could also be explained by a rather unspecific inhibitor, which by definition (being a ATP-competitive inhibitor) will hit MANY kinases. As a side note, the IC50 for this inhibitor (cell-free) is 1 nM!!! Furthermore, they should perform rescues experiments in vitro/ in vivo using HHEX S163E/S177E.

28. Furthermore, Verteporfin is a very "dirty drug" even though (unfortunately) commonly used in the Hippo field.

Minor points:

1. Abstract: "this study, we observed that HHEX in "Please explain what HHEX is beforehand.
2. Fig.1E: Blot IP HA overexposed, shorter exposure time possible?

3. Several other Blots are also overexposed: Fig1I, 5C
4. Fig1G: Efficiency of shRNAs targeting HHEX in this experiment is not traceable. The authors show this only in a separate experiment, but not for the YAP-TEAD Co-IP. Please provide an HHEX Blot in this experiment.
5. Fig1G: Quality of the pan-TEAD Blots is low. On the right blot bands are barely visible. Longer exposure time would be better. Therefore, one can does not state whether HHEX overexpression or knockdown by shRNA has any effects on TEAD expression.
6. Fig2D: Proportions of the VENN diagram misleading, since most of the HHEX dependent genes are independent of TEAD and YAP
7. Fig2H/I: Missing x-axis labeling: Final tumor weight [g].
8. Fig3F: Scale bars are barely visible
9. Fig4D: Scale bars are barely visible
10. Fig4I: Scale bars are barely visible
11. Fig5G: Labeling mistake: Should it be HHEXS163/177C instead of HHEXS163/117C?

Reviewer #2 (Remarks to the Author)

Reviewers comments, revised on submitted manuscript:

Activation of the YAP-TEAD4-HHEX transcriptional complex by Casein Kinase 2
Guo-Y, Zhou-Z, Du-P, Liu-Chen-Ying

The current study by Guo-Y et al. describes a new cooperative regulatory mechanism of YAP/TEAD4 activity by HHEX, in which HHEX plays a positive role as a new interacting partner of the complex. The authors conclude that the interactions of HHEX with both YAP and TEAD are required for full activation of the YAP/TEAD complex, and they present results from HHEX knockdown cells showing downregulation of target genes of YAP/TEAD. It is interesting to note that HHEX-ChIP-qPCR analysis has demonstrated that HHEX binds to the promoter or enhancer regions of the YAP/TEAD target genes. HHEX phosphorylation at S163/S177 by CK2 is suggested to enhance HHEX/TEAD interactions in vivo, which in turn activates YAP. HHEX also supports the progression of colorectal carcinomas. Therefore, it is worth noting that combined treatment with a chemical compound targeting the YAP-TEAD interaction, and CK2i, CX-4945, showed a synergistic inhibitory effect on cell proliferation in CRC cells.

This is a detailed study demonstrating a mechanistic relationship of transcriptional activation. Phosphorylation of HHEX by CK2 positively regulates the interaction of HHEX with TEAD4. These findings are of considerable interest for an area requiring investigation; however, a number of points need to be clarified and some statements require further justification, as discussed below.

1. The section concerning the interaction between HHEX and the YAP/TEAD complex is too repetitive. The formulation and style of some sections, in particular those concerning interactions between YAP and TEAD, which have been enhanced by overexpression of HHEX, require careful review. For example, the authors describe "significantly enhanced... interactions" in lines 94-95, "the F32E mutation in HHEX enhanced its interactions" in lines 114-115, and also in lines 130-131, concerning YAP-TEAD4 interactions, but those differences are not so obvious nor supported statistically by their data. Another source of confusion was caused by some errors in Figure 1E: Are the molecular weights of Flag-HHEX and HA-YAP greater than 70KD and 40KD, respectively? In addition, I could not fully understand the conclusion in Figures S2 C and D, in which the authors claim that YAP could not interact with TLE-1. Why couldn't TLE-1 be in the anti-YAP complexes in the presence of HHEX, which associate with TLE-1 in vivo. Similarly, the difference between WT and the S100A mutation in TEAD4 for the interaction with HHEX does not seem as large as is shown in Figure 3C, although the authors mentioned that a mutation in TEAD4 dramatically diminished its interaction with HHEX. Authors need to be careful, so that the reader can judge the validity of the their findings.

2. The study examines the effects of knockdown of HHEX on xenograft growth and expression levels of YAP/TEAD target genes using HCT-116 cells. HHEX was previously reported as a tumor suppressor in solid tumors; therefore, whether HHEX plays a pro-oncogenic role in colorectal carcinoma cells could be a critical finding. However, the conditions of xenograft tumors derived

from HCT-116 cells are poorly defined: How were HCT-116 cells stably expressing the indicated constructs prepared? How many cells were inoculated, and how long was xenograft growth in mice followed? For the benefit of the reader, the authors should provide detailed descriptions of experimental protocols.

3. It is advisable to provide the sample sizes (N analyzed) for immunohistochemistry and establishment of the colitis-associated CRC mouse model, as shown in Figure 4.

4. The authors' observation that CK2 positively regulates the HHEX/TEAD4 interaction by phosphorylating HHEX demonstrates a new insight for activation of HHEX as a pro-tumorigenic gene. Although the authors stress that serum stimulation promoted formation of the HHEX/TEAD4 complex because of CK2 activated by serum treatment, there is little evidence to support this in the case of HEK-293 cells. The authors need to cite a reference for this statement concerning the CK2 activation in relation to the cell cycle in cultured cells. Additional data are needed to demonstrate the inhibitory effect of CX-4945 or siCK2 during the time course of serum stimulation, as shown in Figure 5D. In spite of S177 in HHEX, S163 does not completely match the phosphorylation site of CK2. Otherwise, the site could run on prime phosphorylation for another site.

5. The Discussion does not adequately elaborate the above points to strengthen the main findings.

Point-by-Point Responses to Reviewers' Comments

Reviewer #1 (Remarks to the Author)

The article by Guo Y. and Zhu Z. et al investigates the cooperation of HHEX with the YAP-TEAD complex and highlights its role in activation of a YAP dependent gene set in colorectal cancer. The authors suggests that HHEX can co-activate a YAP-TEAD driven transcriptional program in a CRC cell line and therefore HHEX possess oncogenic features in CRC. Multifaceted biochemical experiments regarding binding interactions and domain characterizations were carried out and the authors propose that phosphorylation of HHEX by casein kinase 2 (CK2) enhances its binding to TEAD-YAP. Treatment of the CK2 inhibitor, CX-4945, has synergistic effects in CRC when combined with the YAP-TEAD inhibitor verteporfin. However, the mechanistic experiments supporting this mechanism are not very well worked out and are largely correlative. A detailed list is given below. Given the potential important role of HHEX in CRC tumorigenesis, the reviewer recognizes the contribution to the understanding of YAP-TEAD transcriptional control. However, because of the extensive in vivo and in vitro experiments that need to be performed to support their model and conclusions, the reviewer would not recommend this article for a revised version. In case, the authors will hand in a revised version of this manuscript, please provide a reviewer link for their NGS data.

We thank the reviewer for the critical but constructive comments. Following these comments, we performed extensive additional experiments to address the raised issues, which we found truly helpful in strengthening our study and improving the quality of the manuscript. Please note that the NGS data of this study (GSE196333) was released before the submission of this revised manuscript.

The authors should address the issues below:

Major points:

General comments:

1. In most of the experiment's information about the number of replicates are missing, e.g. figure 2A, B, C, F, G; figure 3A, B, C, D, F; figure 4C, D, E, I (ki67 panel); figure 5 E, F, G; figure 6 A, B, E, F. Bar plots should include individual data points like done for Fig4E/F and number of replicates should be included in the figure legend.

Thanks for pointing out this issue. Following the reviewer's comments, we now include the individual data points of the bar plots in the revised manuscript; and the number of replicates of each experiment are now explicitly stated in the figures and figure legends.

2. In all Co-IP experiments, loading controls are missing! CoIP experiments where authors showed enhanced or reduced binding of HHEX to TEAD should be quantified, e.g. Fig 1E, 5A, 5B, 5C

Agree. We now used β -ACTIN in the frozen INPUT samples of the Co-IP experiments as a loading control. We also added quantification of the Co-IP

experiment to indicate change of protein interactions or protein levels in the new Figure 1E-F, 4B, 4G, 5A-D, 6C, 7B, 7C, S2B, S2G, S3B, S3C, S4F, S6C, S6H, S7C, S7F.

3. ChIP-Seq is required to demonstrate the general co-occupancy of TEAD/YAP/TAZ and HHEX.

Following the reviewer's suggestion, we performed ChIP-seq analysis of HHEX in HCT-116 cells by using the HHEX antibody from Abcam (ab34222). After screening three commercial HHEX antibodies, we found that the HHEX antibody from Abcam (ab34222) was suitable for immunoprecipitation. ChIP-qPCR analysis of the CTGF promoter further confirmed that this HHEX antibody could be used for ChIP-seq.

The ChIP-seq data of TEAD4 in HCT-116 cells are available in the ENCODE database and used to analyze the co-occupancy of TEAD4 and HHEX. As shown in new Figure 2H, 16% of TEAD4 target genes are also co-regulated by HHEX in HCT-116 cells ($p=1.19e^{-118}$). The representative co-occupancy of TEAD4 and HHEX at the classical YAP target genes, CTGF, CYR61, LATS2 and AMOTL2 are also included in this revised manuscript (new Figure 2I). Furthermore, we also analyzed the available ChIP-seq data of TEAD4 and GFP-HHEX in HepG2 cells in the ENCODE database, which shows more significant enrichment of HHEX in regions surrounding TEAD4 binding sites (new Figure S4H, S4I).

Specific:

4. Fig. 1C: How did the authors control for DNase activity in this assay? The alternative conclusion here would be that the DNase treatment did not work. Given the involvement of DNA binding domains in the interaction, this point is critical.

Good point. We also asked ourself the same question when we got the negative result at the first time. To rule out the possibility that the DNase treatment didn't work, we analyzed the DNA degradation after the DNase treatment and optimized the experimental condition of DNase treatment. As shown in Figure 1 for reviewer, no DNA was detected after the treatment of Benzonase from Sigma at 37 degree which demonstrated that Benzonase treatment did work well. After Benzonase treatment for 0.5h, the cell lysate was further used for Co-IP at 4 degree as regular.

In addition, under the same condition of Benzonase treatment, we observed the interaction between the transcription factor ELK4 and PARP1 rely on the DNA in another project (Figure 2 for the reviewer), which could be a positive control for the Benzonase treatment.

That said, the involvement of DNA binding domains of both TEAD4 and HHEX in the interaction also puzzled and attracted us. We are working with structural biologists to dissect the interaction interface of TEAD4/HHEX.

5. Fig1E: Can the results from the CoIP studies be quantified to better estimate the effect of HHEX expression on the YAP-TEAD interaction? Also, it would be nice to have a loading control in this experiment, to prove equal loading of the gel.

As the reviewer pointed out earlier, quantification and loading control are required. We now included quantification and loading control in new Figure 1E.

6. Fig. 1F: The authors should remove the model at this position. Here, there could be thousand different ways how HHEX affects TEAD/YAP/TAZ.

Following the reviewer's suggestion, we now removed the model here.

7. Fig. 1G: An immunoblot to demonstrate HHEX KD efficiency/ HHEX oe is absolutely required.

Agree! We now reperformed the experiments with efficiency of HHEX KD/OE examined by immunoblot and quantified.

8. Fig1H: Quality of the PLA assay is rather low. Pictures look like the assay did not work probably, because there are also foci outside the nucleus and a lot of background. In addition, the quality of the controls used in this assay is not appropriate. As this is the only experiment proving endogenous interaction between HHEX-TEAD and HHEX-YAP, this should be crystal clear. Can the authors provide PLA assay using single antibody controls

(either single anti-HHEX or single anti-YAP or single anti TEAD). Preferable would be an PLA assay after knockdown of HHEX to prove that the signal is specific.

We thank the reviewer for the constructive comments to further improve the quality of our study. Following the reviewer's suggestions, we showed that knockdown of HHEX significantly reduced the PLA foci which indicated the PLA signal is specific (new Figure 1H). Additionally, single antibody controls are provided (new Figure S1E).

Both HHEX (Peter Noy, et al., Nucleic Acids Res, 2012, PMID: 22844093) and TEAD4 (PNAS, 2012, PMID: 22529382; Carcinogenesis, 2014, PMID: 24325916; Development, 2018, PMID: 30201685; Nat Cell Biol., 2017, PMID: 28752853) have been reported to exist in the cytoplasm and translocate between cytoplasm and nucleus. We also performed the co-IP assay after the nuclear-cytoplasmic fractions and found that though TEAD4 mainly located in the nucleus, there is a portion of TEAD4 which exists and interacts with HHEX in the cytoplasm (new Figure S6B). Furthermore, knockdown of HHEX also diminished the PLA foci outside the nucleus. Thus, the PLA foci outside the nucleus should be the specific signal of the TEAD4/HHEX.

9. The authors should perform endogenous HHEX TEAD/YAP/TAZ CoIPs.

According to the reviewer's suggestions, we performed the endogenous Co-IP of HHEX/TEAD and HHEX/YAP. Endogenous HHEX was pulled down by both endogenous YAP and TEAD in HCT-116 CRC cells (new Figure 1G), which further supports the interaction between HHEX and YAP/TEAD in CRC cells.

10. Fig. 2A-C: Immunoblots for the transfected proteins are required. What about the F32E mutant that still interacts with TEAD/Y/T? Does this behave as a stronger activator?

Following the reviewer's comments, immunoblots for the overexpressed proteins related with the luciferase assays are now provided in the new Figure 2A-C, 5G, S4A, S4B, S6F in this revised manuscript. HHEX was first reported as a transcriptional repressor by recruiting co-repressor TLE1; and F32E mutation of HHEX disrupts its interaction with TLE1 (J Biol Chem, 2004, PMID:15187083). Our data indicated that different pools of HHEX form protein complexes with YAP and TLE-1 (Figure S2C, S2D), thus the HHEX F32E mutant which is unable to interact with TLE1 would bind more YAP (Figure S2E). Indeed, we found that F32E mutant acted as a stronger activator of YAP as shown by the luciferase assay (new Figure S4B).

11. Fig.2D: The number of differentially regulated genes for a Y/T KD is minimal indicating technical issues. That result is very confusing/concerning. How can HHEX Kd have such a strong effect on YAP-dependent tumor growth even though a minimal number of genes are affected? Here, also immunoblots for all samples are absolutely required. Maybe KD of HHEX also affects TEAD Y/T protein levels. A hypergeometric test is required to demonstrate the significant overlap of the DGEs.

The reviewer raised an important issue. Actually, the original Fig. 2D showed the number of the downregulated (not all differentially expressed) genes after TEADs, YAP/TAZ, HHEX KD in HCT-116 cells. We set a strict bar (FC>2, p<0.05) for the differentially regulated genes and the full list of differentially regulated genes could be found in the Original Supplementary Table S1. We are sorry that this key information was missing during our downsizing the manuscript to meet the word count limit. Following the reviewer’s suggestion, a hypergeometric test was performed (new Figure 2D) and the key information of analyzing the differentially regulated genes has been indicated in the main text and related Figure legend in this revised version. Actually, YAP/TAZ knockdown widely affected the gene expression profiles that 2089 genes’ expression were

downregulated ($p < 0.05$) upon YAP/TAZ knockdown in HCT-116. 23 known YAP target genes were downregulated with $FC > 2$ (CTGF, F3, AMOTL2, CYR61, ANGPT2, CCND1, GPRC5A, AXL, THBS1, FOSL1, SLC2A1, CRIM1, NUAQ2, SOCS3, IRS2, TAGLN, SLC7A5, BCL2L1, LATS2, CAV1, HK2, SLC1A3, TERT), which indicates that our RNA-seq analysis of Y/T KD works well.

As the reviewer pointed out, knockdown of HHEX might also affect protein levels of TEAD, YAP/TAZ. Indeed, we have observed that HHEX KD reduced both the mRNA and protein levels of YAP/TAZ in HCT-116 cells, however, HHEX OE increased the protein levels of YAP/TAZ but didn't affect the mRNA levels of YAP/TAZ (new Figure S4F, S4G). Given the fact that HHEX can affect both gene transcription and mRNA transport (EMBO, 2003, PMID: 12554669), the mechanism of HHEX's effect on YAP/TAZ could be complicated. Thus, we plan to figure it out in our next project, and we discussed this point in the Discussion Section in this revised manuscript.

12. The reviewer asks the authors for GEO access for the data

The NGS data of this study (GSE196333) has been released before the submission of this revised manuscript.

13. Fig. 2E: Here, the labels are missing. The way the GSEA is drawn here, it would demonstrate a significant UPREGULATION of YAP target genes. However, according to their model those genes should be downregulated after HHEX knockdown?

The original Fig. 2E showed a significant upregulation of YAP target genes in the NC/control samples. Yes, a significant downregulation of YAP target gene upon HHEX knockdown was observed and now shown in new Figure 2E in this

revised manuscript.

14. Fig2F: Apparently, the authors performed qPCR analysis on duplicates? If so, this is not sufficient for meaningful statistical analysis. The heat map should depict Z-scores to account for variance.

Three independent siRNA targeting HHEX were used for the RNA-seq analysis of HHEX in HCT-116 cells and two independent HHEX siRNA were routinely used for the further qPCR confirmation and subsequent functional experiments in this study. Thus, for the old Fig. 2F, we just showed the data of two NC and two HHEX kd samples. According to the reviewer's suggestions, the data of the third NC and HHEX kd samples are now included in the new Figure 2F. Following the reviewer's suggestions, we modified the heat map based on the Z-scores.

15. Fig2G: Please include a negative control region in the experiment, only IgG control is not sufficient. Also, it would be interesting to know whether HHEX binding is not affected by TEAD depletion at HHEX "only" genes. Please provide data for that. Again, immunoblots are missing in order to draw meaningful conclusions.

Following the reviewer's suggestions, a negative control region (primers refers to Genes Dev., 2020, PMID: 31831627) and immunoblots showing the TEAD1/4 knockdown and FLAG-HHEX overexpression are provided (new Figure 2J, S4J). In this revised manuscript, we showed that the binding of HHEX on VEGFA genomic locus, the well-known HHEX target gene, is not affected by TEAD kd in HCT-116 cells (new Figure S4K).

16. Fig2H/I also Fig3: The effect of shHHEX on tumor growth looks very convincing. However, here again HHEX immunoblots are absolutely required. Furthermore, this experiment does not demonstrate that the effect of HHEX is YAP-dependent, since HHEX on its own has dramatic effects. The authors should repeat the HHEX oe with the S163/177 phosphomutants (dead and mimetic vs. wt).

In fact, the HHEX immunoblots of the stable HCT-116 (original Fig. S1C) and SW-480 (original Fig. S4D) cells were provided. The HHEX immunoblots related with Fig. 2H/I are now included in the new Figure S5A, S5D. For the Fig. 2H/I, by comparing the YAP/shHHEX with EV/shHHEX or YAP^{5SA}/shHHEX with EV/shHHEX, the tumor suppressive effect of shHHEX were partially reversed by overexpression of WT YAP and growth of shHHEX tumors were fully rescued by overexpression of constitutively active YAP^{5SA}, supporting that the effect of HHEX is YAP-dependent. That said, we agree with the reviewer that the strong oncogenic effect of HHEX is not entirely dependent on YAP. We now discussed this point in the revised manuscript.

According to the reviewer's suggestion, we performed xenograft tumor formation assay using HCT-116 cells stably expressing HHEX WT or CC, EE mutants (new Figure 5J). Consistent with our model, phosphomimetic EE mutant showed enhanced pro-tumorigenic effect compared with the WT HHEX. The CC mutation resulted in loss of pro-tumorigenic effect of HHEX and even decreased the tumor growth of HCT-116 xenografts, indicating a vital role of S163/177 phosphorylation for the pro-tumorigenic function of HHEX.

17. Fig.2/3: Do these tumors really depend on YAP/TAZ for proliferation? Here, also Y/T-independent CRC cell lines are required as ctrl. For this, see the recent Cancer Cell paper by the Bremner lab.

We have read with great interest the Cancer Cell paper by Bremner and colleagues. However, the detail list of YAP/TAZ dependent and independent cell lines are not provided along with the paper. HCT-116 and SW-480 CRC cells are widely used for analyzing the oncogenic function of YAP/TAZ. Knockdown of YAP inhibited cell proliferation in HCT-116 and SW-480 cells (Joseph Rosenbluh, et al., Cell, 2012, PMID: 23245941). In this study, we propose that HHEX can promote the transcriptional activity of the YAP/TEAD complex for tumorigenesis. But, in the Hippo field, various studies have also shown that multiple downstream target genes mediate the oncogenic function of YAP/TAZ in cell proliferation. Thus, we argue that the oncogenic effect of HHEX is not entirely dependent on YAP; instead HHEX may also promote cell proliferation in a YAP/TAZ-independent manner. This is in keeping with the dramatic effect on the gene expression profiles after HHEX knockdown in CRC cells. As mentioned above, we now discussed this point in the revised manuscript.

18. Fig. 3F: To me, these pictures do not look very "representative". E.g. there is not a single Ki-67 positive cell in the shHHEX_1 sample but the quantification shows 20%.

The quantification of Ki-67⁺ cells was performed by using the Image J software with the default parameter. Following the reviewer's comments, we adjusted the gray threshold for the Ki-67 positive cells and re-analyzed the Ki-67 IHC images

(new Figure 3F).

19. Fig. 4A: Is there an effect on Y/T or TEAD protein or Y/T phosphor status?

Following the reviewer's point, we detected the protein levels of YAP/TAZ/TEAD and pSer127-YAP with the protein samples from Hhex KO mice (Figure 3 for reviewer). We observed downregulated protein levels of YAP/TAZ, findings consistent with the observation in the HHEX knockdown CRC cells (new Figure S4F). The p-Ser127 YAP is also slightly increased in the Hhex KO intestine tissues, but the protein level of TEAD4 remains unchanged. Combined with the data in HHEX knockdown CRC cells, HHEX could affect YAP/TAZ in CRC cells. We plan to figure it out in our next project and have discussed this point in the Discussion Section in this revised manuscript.

Figure 3 for reviewer

New Figure S4F

20. Fig5A: The authors show a direct interaction between CK2, HHEX and TEAD. However, in your model (Fig. 5H) CK2 is in the cytoplasm and phosphorylates HHEX. How can it then interact with TEAD4?

The reviewer raised an interesting and important question. To figure out the subcellular site where the TEAD4-CK2/HHEX interaction occurs, the cytoplasmic proteins and nuclear proteins were fractionated for co-IP assays. Compared with CK2 and HHEX, TEAD4 is mainly expressed in the nucleus. Slight expression of TEAD4 was detected in the cytoplasm (new Figure S6B). TEAD4 forms a stable complex with CK2 in the nucleus and overexpression of CK2 can enhance its interaction with HHEX in both cytoplasm and nucleus. In this revised manuscript, we modified the working model to better illustrate the

CK2-HHEX-TEAD4 axis (new Figure 5K).

21. Fig. 5A: Is the increased interaction dependent on the catalytic activity of CK2?

Following the reviewer's suggestions, we explored the effect of CK2α kinase dead mutant on HHEX/TEAD4 interaction. Overexpression of wildtype but not the kinase-dead CK2α promoted the interaction between HHEX and TEAD4 (new Figure 5A). Consistent with the observation that CK2 kinase inhibitor attenuated HHEX/TEAD4 interaction, these results demonstrate the increased HHEX/TEAD4 interaction relies on the kinase activity of CK2.

22. Fig5B: Blot showing CK2 expression is missing in this experiment. Data provided in Sup Fig. 6C is also not convincing since there is almost no difference in the CK2 expression after siCK2 treatment. Also, there is a discrepancy between the knockdown efficiency on

protein and RNA level.

To relieve the reviewer's concern, we now include blot showing CK2 expression upon CK2 knockdown in the new Figure 5B. The decreased interaction between HHEX and TEAD4 was quantified (new Figure 5B, 5C). For the original Fig. S6C, the siCK2 treatment was performed by co-transfection of siRNAs targeting CK2 α and CK2 β , respectively. We detected the decreased protein level of CK2 α by western blot and the quantification of the blot showed an over 50% decrease of CK2 α protein level, which is in keeping with the knockdown efficiency of CK2 α RNA level (around 40%) (new Figure S6H). The CK2 α blot has been replaced with a short exposure in this revised manuscript (new Figure S6H).

23. Fig 5 in general: Effect of the inhibitor CX-4945 as well as siCK2 seem to have only little effects on the HHEX-TEAD interaction (Fig.5B), whereas the interaction is completely abolished when S163/177 are mutated (Fig.5C).

Following the reviewer's suggest, we quantified the blots, which indicated an ~50% decrease of the interaction between HHEX and TEAD4 upon siCK2 or CX-4945 treatment (new Figure 5B). Since neither siCK2 nor the inhibitor CX-4945 could absolutely inhibit CK2 activity, it is conceivable that their blocking effect on HHEX phosphorylation is not as complete as the S163/177C mutation.

24. Do the HHEX mutants S163/177C and S163/177E, siCK2, CX-4945 treatment affect

relative TEAD luciferase activity similar to experiments done in Fig.2?

Good point! Following the reviewer's suggestions, we performed the luciferase assay to explore the effect of the HHEX S163/177 mutation on TEAD luciferase activity. Results showed that co-expression of YAP with HHEX S163/177E but not the S163/177C mutant significantly enhanced TEAD luciferase activity when compared with the WT HHEX (new Figure 5G), though the HHEX S163/177 mutation didn't affect the interaction between HHEX and YAP (new Figure S6E). Meanwhile, both siCK2 and CX-4945 treatment significantly attenuated the HHEX-induced TEAD transactivation (new Figure S6F).

25. After knockdown of HHEX in HCT-116 cells, can the expression of YAP target genes be restored by WT HHEX in comparison to the CC or EE HHEX mutants? One would expect that CC could not restore and EE is doing a better job than WT.

Thanks for this nice suggestion. Accordingly, we detected the mRNA levels of YAP target genes in the siHHEX_3 (targeting the 3'-UTR) HCT-116 cells with

re-expression of WT and mutant HHEX (new Figure 5H, S6J). As the reviewer pointed out, both WT HHEX and EE mutant but not the CC mutant rescued the mRNA levels of CTGF, CYR61 and AXL in the HHEX knockdown cells; and the EE mutant seems to do a better job than WT. Consistently, overexpression of WT and EE mutant of HHEX promoted the tumor growth of HCT-116 xenografts; while overexpression of HHEX CC mutant suppressed tumor growth likely due to its dominant negative effect (new Figure 5J).

New Figure 5H

New Figure S6J

New Figure 5J

26. Does knockdown of CK2 affect YAP expression, stability or localization and therefore the complex of YAP-TEAD-HHEX is less stable?

Following the reviewer's suggestions, we performed additional experiments and found that knockdown of CK2 resulted in decreased mRNA and protein levels of YAP in HCT-116 cells (new Figure S6H). Apparently, CK2 can modulate the activity of YAP-TEAD-HHEX complex through multiple mechanisms. We now discussed this point in this revised manuscript.

New Figure S6H

27. Fig. 6: General comment for the use of the inhibitors. The authors do not provide any evidence, that CX inhibition even does what they propose it does, namely that it hits HHEX S163/177. This they have to convincingly demonstrate. Otherwise, Fig. 6 could also be explained by a rather unspecific inhibitor, which by definition (being a ATP-competitive inhibitor) will hit MANY kinases. As a side note, the IC50 for this inhibitor (cell-free) is 1 nM!! Furthermore, they should perform rescues experiments in vitro/ in vivo using HHEX S163E/S177E.

Thanks for the advice. Due to its high potency, selectivity and safety, CX-4945 (Silmitasertib) is the most widely used CK2 inhibitor. As an orally bioavailable selective inhibitor of CK2, CX-4945 has been used in various clinical trials (clinicaltrials.gov). As the reviewer mentioned, the IC50 of CX-4945 in cell free assay is only 1 nM and only 7 of the 238 tested kinases were inhibited by CX-4945 in cell free assay with the IC50 range from 17 to 56 nM (Cancer Res., 2010, PMID: 21159648). CX-4945 (10 μ M) didn't affect most of the off-target kinases in the cell-based assays, which further indicates the high selectivity of CX-4945 for inhibiting CK2 (Cancer Res., 2010, PMID: 21159648). Besides, according to the literatures, the EC50 of CX-4945 in cell-based assays ranged from 1 to 20 μ M (the full table can be found in the Selleck product website:https://www.selleckchem.com/products/cx-4945-silmitasertib.html#s_ref).

Following the reviewer's suggestions, we further detected the phosphorylation of HHEX by a pan-phospho-Ser/Thr antibody. We found that CX-4945 treatment decreased the phosphorylation level of WT HHEX but didn't affect the EE mutant. These results indicate that CX-4945 indeed targets HHEX S163/177 (new Figure S6C), findings consistent with the previous report (Nucleic Acids Res., 2009, PMID: 19324893). Moreover, as the reviewer suggested, we performed the rescue experiments and tested the inhibitory effect of single agents or combination treatment in the HCT-116 cells stably expressed WT/EE HHEX. The synergistic effect of CX-4945 and Super-TDU was abolished in cells overexpressing EE mutant but not WT HHEX (new Figure S7D-F). These data further support that CX-4945 indeed hits HHEX S163/177.

28. Furthermore, Verteporfin is a very "dirty drug" even though (unfortunately) commonly used in the Hippo field.

The reviewer is correct about the "dirty drug" and its situation in the Hippo field. Verteporfin (VP) is the first tool compound to disrupt YAP/TEAD complex, which shows a reasonable on-target effect in terms of YAP/TEAD activity. Given its arguably the best on-target effect as a tool inhibitor available, VP is commonly used in the Hippo field. We do agree with the reviewer that VP is a "dirty drug". In our previous study, we have developed an inhibitory peptide disrupting YAP-TEAD interaction based on the VGLL4-TEAD interaction (Cancer Cell, 2014, PMID: 24525233). Similar to the combination treatment with CX-4945 and VP, we tested the synergistic effect of CX-4945 and Super-TDU in three CRC cell lines. We reproducibly observed that CX-4945 and Super-TDU synergistically inhibited the cell viability of CRC cells (new Figure S7A-C).

New Figure S7C

Minor points:

1. Abstract: "this study, we observed that HHEX in "Please explain what HHEX is beforehand.

Many thanks for reminding us this issue. We now introduced HHEX accordingly.

2. Fig.1E: Blot IP HA overexposed; shorter exposure time possible?

Done. We repeated the HA blots with a shorter exposure (new Figure 1E).

New Figure 1E

3. Several other Blots are also overexposed: Fig1I, 5C

We repeated the assay with a shorter exposure for Fig1I and the overexposed IP HA blots in old Fig5C have been replaced with shorter exposure of IP HA images in this revised manuscript (new Figure 1I, 5C).

New Figure 1I

New Figure 5C

4. Fig1G: Efficiency of shRNAs targeting HHEX in this experiment is not traceable. The authors show this only in a separate experiment, but not for the YAP-TEAD Co-IP. Please provide an HHEX Blot in this experiment.

Following the reviewer's comments, we repeated the co-IP experiments with HHEX/ β -ACTIN blots included (new Figure 1F).

5. Fig1G: Quality of the pan-TEAD Blots is low. On the right blot bands are barely visible. Longer exposure time would be better. Therefore, one can not state whether HHEX overexpression or knockdown by shRNA has any effects on TEAD expression.

We repeated the co-IP experiments and the quality of the TEAD blots are improved in the new Figure 1F. We didn't observe any effect of HHEX on TEAD expression at both mRNA and protein levels in this study.

6. Fig2D: Proportions of the VENN diagram misleading, since most of the HHEX dependent genes are independent of TEAD and YAP

Agree. We modified the Figure 2D as the reviewer suggested.

7. Fig2H/I: Missing x-axis labeling: Final tumor weight [g].

Many thanks for pointing this out. We now corrected the figures as the following.

8. Fig3F: Scale bars are barely visible

We modified the figure as the following.

9. Fig4D: Scale bars are barely visible

We modified the figure as the following.

10. Fig4I: Scale bars are barely visible

We modified the figure as the following.

11. Fig5G: Labeling mistake: Should it be HHEXS163/177C instead of HHEXS163/117C?
Many thanks for pointing this out. We now corrected the error in the new Figure 5I.

Reviewer #2 (Remarks to the Author)

The current study by Guo-Y et al. describes a new cooperative regulatory mechanism of YAP/TEAD4 activity by HHEX, in which HHEX plays a positive role as a new interacting partner of the complex. The authors conclude that the interactions of HHEX with both YAP and TEAD are required for full activation of the YAP/TEAD complex, and they present results from HHEX knockdown cells showing downregulation of target genes of YAP/TEAD. It is interesting to note that HHEX-ChIP-qPCR analysis has demonstrated that HHEX binds to the promoter or enhancer regions of the YAP/TEAD target genes. HHEX phosphorylation at S163/S177 by CK2 is suggested to enhance HHEX/TEAD interactions in vivo, which in turn activates YAP. HHEX also supports the progression of colorectal carcinomas. Therefore, it is worth noting that combined treatment with a chemical compound targeting the YAP-TEAD interaction, and CK2i, CX-4945, showed a synergistic inhibitory effect on cell proliferation in CRC cells.

This is a detailed study demonstrating a mechanistic relationship of transcriptional activation. Phosphorylation of HHEX by CK2 positively regulates the interaction of HHEX with TEAD4. These findings are of considerable interest for an area requiring investigation; however, a number of points need to be clarified and some statements require further justification, as discussed below.

We highly appreciate the reviewer's encouraging and constructive comments.

1. The section concerning the interaction between HHEX and the YAP/TEAD complex is too repetitive. The formulation and style of some sections, in particular those concerning interactions between YAP and TEAD, which have been enhanced by overexpression of HHEX, require careful review. For example, the authors describe "significantly enhanced... interactions" in lines 94-95, "the F32E mutation in HHEX enhanced its interactions" in lines 114-115, and also in lines 130-131, concerning YAP-TEAD4 interactions, but those differences are not so obvious nor supported statistically by their data. Another source of confusion was caused by some errors in Figure 1E: Are the molecular weights of Flag-HHEX and HA-YAP greater than 70KD and 40KD, respectively? In addition, I could not fully understand the conclusion in Figures S2 C and D, in which the authors claim that YAP could not interact with TLE-1. Why couldn't TLE-1 be in the anti-YAP complexes in the presence of HHEX, which associate with TLE-1 in vivo. Similarly, the difference between WT and the S100A mutation in TEAD4 for the interaction with HHEX does not seem as large as is shown in Figure 3C, although the authors mentioned that a mutation in TEAD4 dramatically diminished its interaction with HHEX. Authors need to be careful, so that the reader can judge the validity of their findings.

We thank the reviewer for pointing out these issues. To better compare the differences of the protein interactions, we quantified the results of the co-IP experiments (new Figure 1E-F, 4B, 4G, 5A-D, 6C, 7B, 7C, S2B, S2G, S3B, S3C, S4F, S6C, S6H, S7C, S7F). Accordingly, we have modified our wording and statement to describe these data more accurately.

We apologize for the confusion due to errors in the Figure 1E. The bands of

Flag-HHEX are above 40KD and HA-YAP are above 70KD. This error has been corrected in this revised manuscript.

The co-IP data in Figure S2C/D showed that YAP could interact with HHEX but not TLE-1. Thus, we speculated that HHEX forms distinct protein complexes with YAP and TLE-1, respectively. As such, even though YAP doesn't compete with TLE-1 to interact with HHEX, other components in the YAP/HHEX complex could prevent the association of TLE-1 to HHEX in the YAP/HHEX complex.

2. The study examines the effects of knockdown of HHEX on xenograft growth and expression levels of YAP/TEAD target genes using HCT-116 cells. HHEX was previously reported as a tumor suppressor in solid tumors; therefore, whether HHEX plays a pro-oncogenic role in colorectal carcinoma cells could be a critical finding. However, the conditions of xenograft tumors derived from HCT-116 cells are poorly defined: How were HCT-116 cells stably expressing the indicated constructs prepared? How many cells were inoculated, and how long was xenograft growth in mice followed? For the benefit of the reader, the authors should provide detailed descriptions of experimental protocols.

Agree. Following the reviewer's comments, we now briefly described the experimental conditions of xenograft tumors in the Figure legends, and included the detailed experimental procedures in the Methods section as well.

“Cell Lines and Reagents

...For establishing the stable cells overexpressing HHEX, the HHEX cDNA was first cloned into pLVX-puro lentiviral vector. Additionally, FLAG-HHEX WT, S163/177C and S163/177E mutants were amplified by PCR with the 5'-primers containing the FLAG-tag sequence, then cloned into pLVX-puro lentiviral vector. shHHEX-1, shHHEX-2 were generated by using the pLKO.1 vector. The primers and shRNA sequences used in this study are listed in Supplementary Table S3. The retrovirus and lentivirus were generated as described in the previous study. Cells were infected with the virus supernatant in the presence of 8 µg/ml polybrene for 24 h. Then stable cells were selected with treatment

with 2 µg/ml puromycin for one week or 200 µg/ml hygromycin for two weeks before subsequential assays.”

“Mouse xenograft experiment

All mouse studies were approved and all animals were manipulated according to the protocols approved by the animal care and use committees of Xinhua Hospital. Male BALB/c nude mice of approximately 4-6 weeks of age were purchased from GemPharmatech, Shanghai, China. Mice were subcutaneously injected with 1×10^6 cells of indicated stable CRC cell lines resuspended in 100 µl phosphate-buffered saline (PBS) into the right flank (n=6 mice per group) and were sacrificed on day 21. For the experiments assessing the xenografts overexpressing YAP^{5SA} and HHEX, the mice were sacrificed on day 18 to avoid excess tumor burden. For drug treatment, after the volume of xenografts reached approximately 100 mm³, the mice were randomly assigned into the indicated groups and the mice were administrated with DMSO vehicle, CX-4945 (25 mg/kg, oral gavage twice daily), VP (25 mg/kg, intraperitoneal injection every other day) or a drug combination in which each compound was administrated at the same dose and scheduled as single agent for three weeks (n=6 mice per group). The tumor volume based on caliper measurements was calculated by the formula: $0.5 \times (\text{largest diameter}) \times (\text{smallest diameter})^2$. At the experimental endpoint, the mice were sacrificed, and the xenografts were dissected, photographed and weighted. Then the xenografts were fixed in 4% paraformaldehyde overnight at room temperature before paraffin embedding and sectioning for IHC analysis.”

3. It is advisable to provide the sample sizes (N analyzed) for immunohistochemistry and establishment of the colitis-associated CRC mouse model, as shown in Figure 4.

Good point. According to the reviewer’s suggestions, the sample sizes information with individual data points in Figure 4 are shown in the new Figure 4F and Figure 4I and also described in the related Figure legends.

4. The authors' observation that CK2 positively regulates the HHEX/TEAD4 interaction by phosphorylating HHEX demonstrates a new insight for activation of HHEX as a pro-tumorigenic gene. Although the authors stress that serum stimulation promoted formation of the HHEX/TEAD4 complex because of CK2 activated by serum treatment, there is little evidence to support this in the case of HEK-293 cells. The authors need to cite a reference for this statement concerning the CK2 activation in relation to the cell cycle in cultured cells. Additional data are needed to demonstrate the inhibitory effect of CX-4945 or siCK2 during the time course of serum stimulation, as shown in Figure 5D. In spite of S177 in HHEX, S163 does not completely match the phosphorylation site of CK2. Otherwise, the site could run on prime phosphorylation for another site.

Following the reviewer's suggestions, we cited two original papers (Serum-stimulated cell growth causes oscillations in casein kinase II activity. J Biol Chem. 1989 ; Protein kinase CK2alpha' is induced by serum as a delayed early gene and cooperates with Ha-ras in fibroblast transformation. J Biol Chem. 1998) which reported the elevation of CK2 activity by serum treatment.

To further relieve the reviewer concern, we tested a possible inhibitory effect of CX-4945 on the serum induced formation of the TEAD4/HHEX complex. As expected, treatment with CK2 inhibitor almost totally blocked the formation of the TEAD4/HHEX complex during the time course of serum stimulation (new Figure 5D). We agree with the reviewer that only S163 (160- KYLSPPE-166) match the consensus motif of CK2 phosphorylation site. In the Abdenour Soufi, et al.'s original paper (Nucleic Acids Res., 2009, PMID: 19324893), a phosphor-peptide with phosphorylation of S163 and S177 were identified by mass spectrometry. Further *in vitro* phosphorylation assay by CK2 showed that both S163E and S177E retained but with reduced pan-phosphorylation signal, and the S163/177E double mutant lost most of the pan-phosphorylation signal. Their data indicated S163 and S177 as the primary phosphorylation sites by CK2.

New Figure 5D

5. The Discussion does not adequately elaborate the above points to strengthen the main findings.

Many thanks for pointing this out. We carefully revised the Discussion section to better elaborate and strengthen our major findings. In particular, we added the dynamic formation of HHEX/TEAD/YAP complex induced by serum and HHEX's potential effect on YAP/TAZ levels.

“It is worth noting that serum leads to a rapid decrease in YAP phosphorylation in minutes which can be recovered after several hours due to the negative feedback loop of Hippo pathway. In contrast, the effect of serum on interaction between HHEX and TEAD4 occurs after the change of YAP phosphorylation and can last longer time which may account for the long-term maintenance of YAP/TEAD activity induced by serum.”

“Besides, our study reveals that knockdown of HHEX induced downregulated mRNA levels of YAP/TAZ but overexpression of HHEX led to upregulation of YAP/TAZ protein levels without change of mRNA levels in CRC cells. Similarly, our study also implicates CK2 could promote gene transcription of YAP/TAZ in CRC, which further supports the enhancing effect of CK2 on YAP/TEAD activity and the pro-tumorigenic function of CK2 in CRC. Since HHEX has been reported to bind with eIF4E and inhibit eIF4E-dependent mRNA transport, we hypothesize that HHEX could regulate YAP/TAZ at both transcriptional and post-transcriptional level in CRC cells and CK2 might regulate YAP/TAZ through HHEX. Furthermore, given the multiple biochemical function of HHEX and the more extensive effect on gene expression profiles by HHEX knockdown compared with YAP in CRC cells based on our RNA-seq analysis, we consider that the strong oncogenic effect of HHEX is not entirely dependent on YAP. Thus, further studies are needed to fully elucidate the mechanism of oncogenic function of HHEX and CK2 in CRC in the future.”

REVIEWER COMMENTS

Reviewer #1 (Remarks to the Author):

The authors now substantially revised the manuscript. Remarkably, the 28 major points were all addressed and every single experiment came out so that it would support their conclusions from the first draft.

Still, a major conceptual concern remains: namely that many of the effects that the authors describe are indirect because shHHEX leads to a strong downregulation of YAP/TAZ on protein level (now provided as Figure S4F). BTW (related to S4F): a pLKO.1 (empty vector) as control for an shRNA experiment is not an appropriate control since overexpressed shRNAs compete for endogenous miRNAs - something that is not captured if an empty vector is used as control.

Kindly, the authors now provided also the requested information on the statistics. Here, however, many shortcomings become unfortunately apparent:

- a $p=0$ is presented in S4I and Fig. 7G. Something like this does not exist!

- Nearly throughout the manuscript, the authors use a Student's T-test, even when comparing > 2 experimental groups. This has to be replaced by an appropriate statistical test including a correction for multiple testing. They even use a Student's T-test on discrete data (!), e.g. an IHC score in Fig. 7D.

I strongly urge the authors to get someone with a background in statistics on board in order to correct these mistakes.

Where a T-test would have been appropriate (normally distributed, 2 samples), they should rather use a Welch test.

Reviewer #2 (Remarks to the Author):

The authors have responded to all questions and addressed all of the reviewers' original concerns. Therefore, I would suggest the revised manuscript is now suitable for publication. I hope that the publication of this paper will attract a wide readership and encourage a broader approach in the field of cancer research.

Point-by-Point Responses to Reviewers' Comments

Reviewer #1 (Remarks to the Author):

The authors now substantially revised the manuscript. Remarkably, the 28 major points were all addressed and every single experiment came out so that it would support their conclusions from the first draft.

We thank the reviewer for the helpful comments during our revision of the manuscript.

Still, a major conceptual concern remains: namely that many of the effects that the authors describe are indirect because shHHEX leads to a strong downregulation of YAP/TAZ on protein level (now provided as Figure S4F).

We agree with the reviewer that HHEX could affect the activity of YAP/TAZ/TEAD complex through both direct and indirect mechanisms. In CRC cells, knockdown of endogenous HHEX could lead to both decreased protein levels of YAP/TAZ and destabilization of YAP/TEAD complex, eventually suppressing CRC cell proliferation and tumor growth (Fig. 3A-F). In these shHHEX models, both direct and indirect mechanisms are involved in HHEX's effect on YAP/TAZ/TEAD activity and pro-tumorigenic function. However, in the YAP^{5SA} overexpression models, the constitutively active YAP^{5SA}/TAZ^{4SA} may dominate and overrule the endogenous YAP/TAZ (increased phosphorylation, cytoplasmic translocation and decreased protein levels of endogenous YAP/TAZ) through activating the negative feedback of Hippo pathway (PMID: 26109050; PMID: 26109051; PMID: 26315483), and therefore in this case, the indirect effect of shHHEX on the YAP^{5SA}/TEAD complex should be minimal. In the TEAD reporter luciferase assay, coexpression of HHEX still significantly promoted the transactivation of TEAD by both YAP^{5SA} and TAZ^{4SA} (Fig. 2A, S4A). Knockdown of HHEX also dramatically suppressed the upregulated transcription of YAP target genes by YAP^{5SA} and attenuated the tumor growth of YAP^{5SA} expressing xenografts (Fig. 2L, S5D-F). In addition, we observed that overexpression of WT and EE mutant HHEX led to similar upregulation of YAP/TAZ protein levels (shown below). Thus, we consider that the enhanced effect of EE mutant HHEX on YAP could be mainly through complexing with YAP/TEAD (Fig. 5H, 5J, S6J). These observations indicated a direct role of HHEX with YAP-TEAD4.

That said, we agree. HHEX can also indirectly enhance the protein levels of YAP/TAZ. Elevated expression of HHEX is correlated with hyperactivation of YAP/TEAD in CRC and associated with poor prognosis in CRC patients. Following the reviewer's comment, we now toned down the direct role of HHEX with YAP-TEAD4 and explicitly discussed the

indirect role of HHEX in our revised manuscript. For example, the indirect effect of HHEX on YAP/TAZ at protein level is included in the Abstract in this revised manuscript.

BTW (related to S4F): a pLKO.1 (empty vector) as control for an shRNA experiment is not an appropriate control since overexpressed shRNAs compete for endogenous miRNAs - something that is not captured if an empty vector is used as control.

As the reviewer pointed out, pLKO.1-Scramble is a better control than pLKO.1-vector. In an alternative experiment, we used scramble siRNA as a control. Similar to the result of Fig. S4F, siHHEX also led to decreased protein levels of YAP/TAZ in HCT116 cells, in which the scramble siRNA was used as a control (shown below).

Kindly, the authors now provided also the requested information on the statistics. Here, however, many shortcomings become unfortunately apparent:

- a p=0 is presented in S4I and Fig. 7G. Something like this does not exist!

Agree! P value can be very small but not be zero. Thanks for pointing this out. For the Fig. S4I, the hypergeometric test was performed by R to assess the statistical significance. If the p value is less than $1e-325$, “p=0” will be returned by R software (PMID: 34718768; PMID: 34928380). We now corrected the p value in the new Fig. S4I ($p < 1e-325$). For the Fig. 7G, the correlation between mRNA level of HHEX and the YAP target gene signature was performed by the GEPIA2 database (<http://gepia2.cancer-pku.cn/>). We have checked the source code of Gepia2 database (<http://gepia2.cancer-pku.cn/#example>). If the p value is less than $2.2e-16$, Gepia2 will return “p=0” as the output for the correlation analysis. The p value in the new Fig. 7G and related supplemental Fig. S8B has been corrected as “ $p < 2.2e-16$ ”.

- Nearly throughout the manuscript, the authors use a Student's T-test, even when comparing > 2 experimental groups. This has to be replaced by an appropriate statistical test including a correction for multiple testing. They even use a Student's T-test on discrete data (!), e.g. an IHC score in Fig. 7D.

I strongly urge the authors to get someone with a background in statistics on board in order to correct these mistakes.

We thank the reviewer for the constructive comments to further improve the quality of our study. We now used one-way ANOVA with Dunnett's multiple comparison test to assess the statistical significance for the experiments with >2 independent groups (Fig. 2A-C, 2J-

L, 3E-F, 4C, 4F, 4H-I, 5F-J, 6B, 6D-F, 7C, S1C, S4A-E, S4G, S4K, S5B-C, S5E-F, S6F, S6J, S7B, S7E). For the CCK8 and xenografts growth curve assays (Fig. 3A, 3C, 6D), two-way ANOVA with Dunnett's multiple comparison test was performed to assess the statistical significance. For the paired IHC score data in Fig. 7D, Wilcoxon signed rank test was used. The correction of the statistical test didn't change the statistical significance of the data, except for the Fig. 2A and S4A that coexpression of HHEX at low level didn't significantly increase the luciferase activity of TEAD reporter. Still, moderate to high levels of HHEX coexpression statistically promoted the transactivation of TEAD (new Fig. 2A, S4A).

Where a T-test would have been appropriate (normally distributed, 2 samples), they should rather use a Welch test.

Following the reviewer's suggestions, a Welch's t-test was performed for the experiments with 2 independent samples (normally distributed data).

REVIEWERS' COMMENTS

Reviewer #1 (Remarks to the Author):

The authors addressed all my concerns regarding statistics.